# Deconstructing the geography of human impacts on species' natural distribution

Conor Waldock [1,2,3] ✉, Bernhard Wegscheider[1,2,3], Dario Josi [1,2,3], Bárbara Borges Calegari[1,2,3,4], Jakob Brodersen [1,2], Luiz Jardim de Queiroz[1,2,5,6] & Ole Seehausen[1,2]

It remains unknown how species' populations across their geographic range are constrained by multiple coincident natural and anthropogenic environmental gradients. Conservation actions are likely undermined without this knowledge because the relative importance of the multiple anthropogenic threats is not set within the context of the natural determinants of species' distributions. We introduce the concept of a species '*shadow distribution*' to address this knowledge gap, using explainable artificial intelligence to deconstruct the environmental building blocks of current species distributions. We assess shadow distributions for multiple threatened freshwater fishes in Switzerland which indicated how and where species respond negatively to threats – with negative threat impacts covering 88% of locations inside species' environmental niches leading to a 25% reduction in environmental suitability. Our findings highlight that conservation of species' geographic distributions is likely insufficient when biodiversity mapping is based on species distribution models, or threat mapping, without also quantifying species' expected or shadow distributions. Overall, we show how priority actions for nature's recovery can be identified and contextualised within the multiple natural constraints on biodiversity to better meet national and international biodiversity targets.

Tackling biodiversity loss remains a major challenge for conservation[1], with surging extinction rates driven by catastrophic declines of abundance within some populations[2,3] and shrinking of species' geographic ranges[4–6]. Alleviating specific threats with targeted and evidence-based conservation actions can be highly beneficial for local species' populations[7–9]. Targeted conservation actions would be most efficient when jointly understanding where species naturally occur, which areas are threatened, and whether species are sensitive to threats. This requires identifying how multiple natural and anthropogenic factors together contribute to structuring species' geographic distributions[10].

At present, most studies focus on revealing the overall ranking of global threats, rather than the local contribution of each threat. For example, identifying land-use change as the main global cause of terrestrial and freshwater biodiversity loss and over-harvesting as the main global cause of marine biodiversity loss[11]. However, the local contribution of each threat to the state of species populations could differ substantially from the global average. As such, effective conservation efforts should alleviate the most limiting anthropogenic threat(s) in each population's location across a species' distribution. Furthermore, conservation planning must account for the broader

[1]Aquatic Ecology and Evolution, Institute of Ecology and Evolution, University of Bern, Bern, Switzerland. [2]Department of Fish Ecology and Evolution, EAWAG, Swiss Federal Institute for Aquatic Science and Technology, Kastanienbaum, Switzerland. [3]Wyss Academy for Nature at the University of Bern, Bern, Switzerland. [4]Department of Vertebrate Zoology, National Museum of Natural History, Smithsonian Institution, Washington, DC, United States of America. [5]Naturalis Biodiversity Center, Leiden, The Netherlands. [6]Groningen Institute for Evolutionary Life Sciences, University of Groningen, Groningen, The Netherlands. ✉e-mail: conor.waldock@unibe.ch; conorwaldock@gmail.com

non-anthropogenic environmental and ecological context alongside threats, and weak management outcomes for biodiversity occur if this broader context is overlooked[12,13]. Implementing conservation actions at local and sub-national scales is also key to achieving international biodiversity targets such as the Kunming-Montreal Global Biodiversity Framework (GBF)[14]. For instance, Target 2 of the GBF calls for 30% of degraded areas to be restored by 2030. However, we often do not know where and which conservation actions will be most effective because the local contribution of each threat to the state of species populations is unknown − limiting the capacity to implement and downscale conservation actions to meet biodiversity targets effectively.

Niche-based theories on the geography of species populations recognise the high dimensionality of species niches[15-18], and a long-standing challenge in ecology is to unravel this dimensionality and reveal where species are constrained by different environmental factors[15]. While ecological niche models and species distribution models (SDMs) are widely applied to make predictions of geographic areas suitable for species populations to occur[19,20], they are less often applied to understand the geography of environmental factors affecting populations. Current research also rarely disentangles the relative influence of natural versus anthropogenic factors on spatial patterns in environmental suitability predictions[21,22], as highlighted above as critical to conservation and restoration planning[13]. Recent work shows that the spatial contribution of different environmental factors can be identified by applying explainable artificial intelligence (XAI) to SDMs[23,24], which highlights new avenues for generating fundamental and applied insights on the geography of environmental constraints on species populations.

Here, we investigate freshwater fish communities, which are the most speciose vertebrate group having an estimated extinction rate of 100x natural rates of extinction[25]. In addition, freshwater fishes live under multiple coincident and spatially structured threats[26-28] that occur along strong natural gradients in dendritic networks[29]. Freshwater fish are, therefore, a good model to achieve the overarching aim of this work: to quantify the relative contribution of multiple environmental factors affecting species' populations in each location across portions of their geographic distributions. We apply XAI (SHapley Additive exPlanations, or SHAP values[30]) combined with species distribution models to estimate the relative contributions of natural factors and anthropogenic threats to local predictions of environmental suitability. SHAP values enable us to decompose net environmental suitability scores for each location into separate contributions from each environmental variable. This provides further observation-level insights compared to traditional variable importance approaches that only show which variables are overall most important for model performance. We also aggregate SHAP values to explore the rarely addressed question of how the positive effects of natural factors (a species' abiotic niche) and the negative effects of threats influence species' populations across their geographic range.

In answering the above, we coin a distributional concept, the 'shadow distribution' of a species (Fig. 1). A shadow distribution is the area where natural abiotic factors defining the realised niche of a species positively contribute to species population performance, but threats, contributing negatively, reduce population performance − quantifying the extent that an observed species distribution is in the shadow of human influence. We also define the 'expected distribution' as all areas where abiotic realised niche factors positively contribute to environmental suitability. We then examined the extent to which shadow distributions mask areas of potentially suitable habitat, which would indicate that using environmental suitability predictions from SDMs greatly underestimates the expected distribution of species. If using indicators of species distributions from environmental suitability predictions alone (e.g.,[31,32] and as indicators for GBF targets 1–3), such differences between raw environmental suitability and expected distributions could undermine the monitoring, implementation, and priority setting of any spatial biodiversity assessment.

For specific locations, we ask, how do multiple environmental factors contribute both negatively and positively to the multiple potential species that could occur in a location? Further, does the relative contribution of these multiple environmental factors vary between contrasting locations? To demonstrate the broad-ranging applications of our approach we provide single- and multi-species assessments across species geographic distributions, in addition to multi-species assessments at specific locations. Overall, our analysis considered eight native species that are threatened or ecologically important and one non-native species, in Switzerland, as well as their responses to 11 natural factors and threats in each of approximately 15,000 river sub-catchments (i.e., around 1.5 million potential local relative contribution scores). We show how quantifying the fundamental natural constraints on species geographic ranges, as well as how threats act within these fundamental constraints, builds better expectations for the spatial distribution of biodiversity to manage the most important threats that constrain realised biodiversity locally.

## Results

### Mapping local relative contributions for single- and multi-species comparisons

The spatial distribution of *Alburnoides bipunctatus* exhibited the highest occurrence in the lower elevation main stem of the Aare River and the adjoining tributaries (Fig. 2a and see Supplementary Fig. 8 for all species). River discharge, connectivity, and temperature were the most important environmental factors contributing to environmental suitability (Fig. 2b, c). The remaining factors of urbanisation, river morphological modification index, distance to lakes, floodplain availability and flow velocity had lower contributions. An overview of the magnitude and direction of all variable effects across species is shown in Fig. 3. Investigating the spatial distribution of SHAP values revealed independent contributions of variables to the spatial distribution of environmental suitability of *A. bipunctatus* (Fig. 2e−l, and across all species Fig. 3 and Supplementary Figs. 9–16). For example, river discharge and connectivity had spatially independent contributions to environmental suitability, even though these variables had similarly high overall importance (Spearman's rank correlation, $\rho = -0.06$). We found low spatial correlations between all pairwise comparisons of SHAP values across all variables for *A. bipunctatus* (|median Spearman's rank correlation| = 0.13, |mean| = 0.2 ± 0.19). The independence between SHAP values for different variables was generated by initial differences in the spatial variation in the raw environmental factors combined with the different effects of each environmental factor on the environmental suitability of *A. bipunctatus* in terms of magnitude, direction, and response shape (i.e., different sensitivity to different factors; insets in Fig. 2e−l and Supplementary Figs. 9–17).

### Deconstructing distributions to build conservation expectations

Overall, a wide range of the areas within the natural realised abiotic niche of *A. bipunctatus* had negative SHAP values for threats, in other words, *A. bipunctatus* had a large shadow distribution (Fig. 4). First partitioning the SHAP values to identify areas where natural factors support *A. bipunctatus* revealed a high percentage of sub-catchments had positive contributions of discharge (33%), minimum temperature (69%), flow velocity (62%) and proximity to lakes (55%) to environmental suitability. However, the area of geographic distribution within the abiotic niche dropped dramatically when we considered all natural factors together in contrast to independently, and this 'expected distribution' covered only 38% of all sub-catchments (i.e., defined as sub-catchments with positive contribution across the sum of natural abiotic SHAP values; Fig. 4a, b). Furthermore, only 6% of all sub-

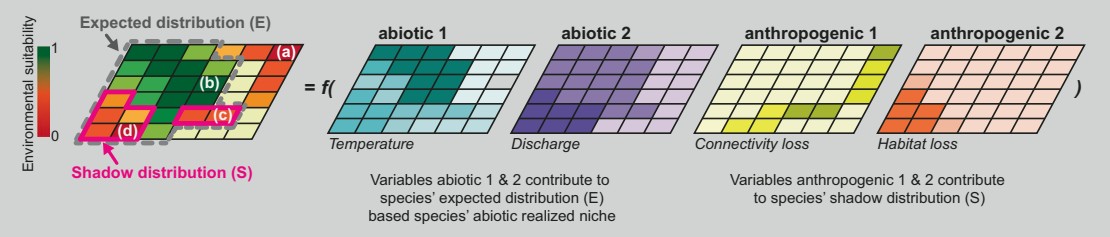

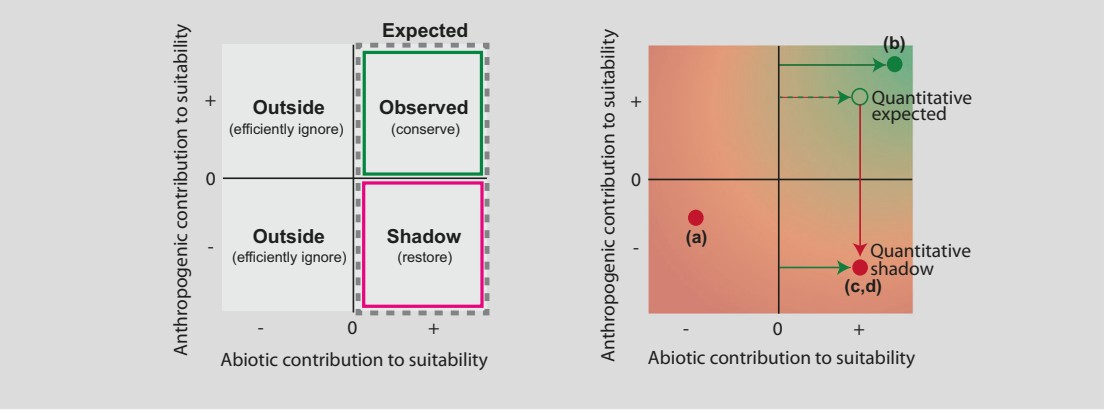

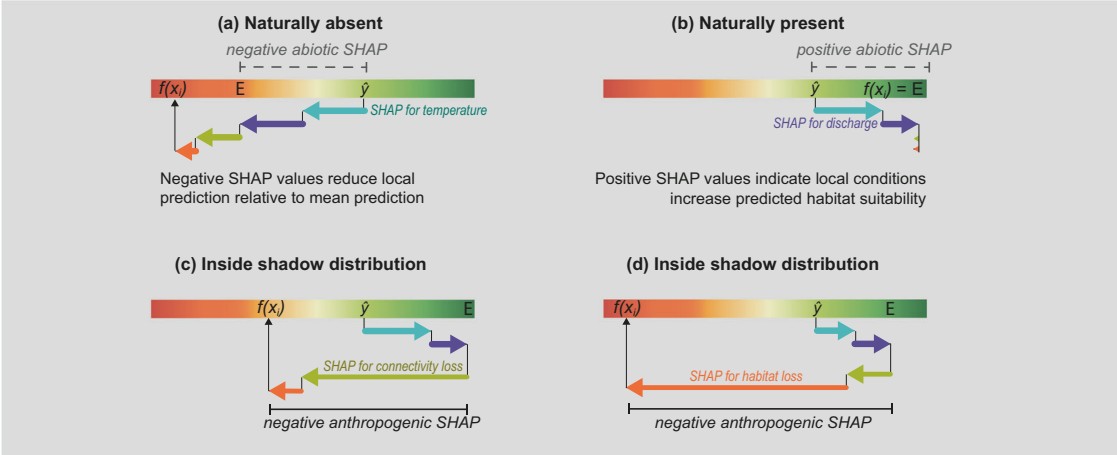

**Fig. 1 | Conceptual definition of species shadow and expected distribution and an overview of our XAI workflow applying SHAP values to species distribution models.** The top panel (**1**) shows the spatial relationship of an ecological property (e.g., species occurrence, presence-absence, or abundance) across a geographic domain depending on various environmental variables (with environmental suitability indicated from red to green). Locations labelled a-d represent contrasting locations within a species' geographic range, each with a different explanation for the local environmental suitability scores $y_i$ defined by model function $f(x_i)$. The locations inside the expected distributions (*E*) and shadow distribution (*S*) are indicated by coloured borders. The middle panel (**2**) shows a theoretical biplot of species distributions depending on abiotic and anthropogenic contributions to environmental suitability. The binary expected distribution is defined where abiotic factors have positive SHAP values (grey dashed lines), while the observed distribution is where both abiotic and anthropogenic contributions are positive. This observed distribution is typically identified by SDMs that do not account for anthropogenic effects. The shadow distribution comprises areas where abiotic factors have positive SHAP values, but anthropogenic factors have negative SHAP values, as defined by Eqs. 1–4. Conservation strategies are suggested for each distribution. Quantitative definitions on the right of the panel (**2**) indicate that locations c and d fall within the observed distribution, but anthropogenic impacts reduce the expected distribution to the shadow distribution. Lower panel (**3**) shows how SHAP values categorise areas as inside or outside of shadow and expected distributions. a shows a site where negative SHAP values for natural abiotic factors reduced environmental suitability from the expected value (*E*), indicating it is outside of the species' expected distribution. b shows a site with a positive SHAP value for natural abiotic factors, indicating the species is naturally present. c, d represent a site where the positive effects of natural abiotic factors are countered by the negative effects of anthropogenic factors placing the site in the 'shadow distribution'. Notably, (a) and (b) fall outside of the shadow distribution because (a) is not inside the expected distribution and (b) does not have negative anthropogenic SHAP values.

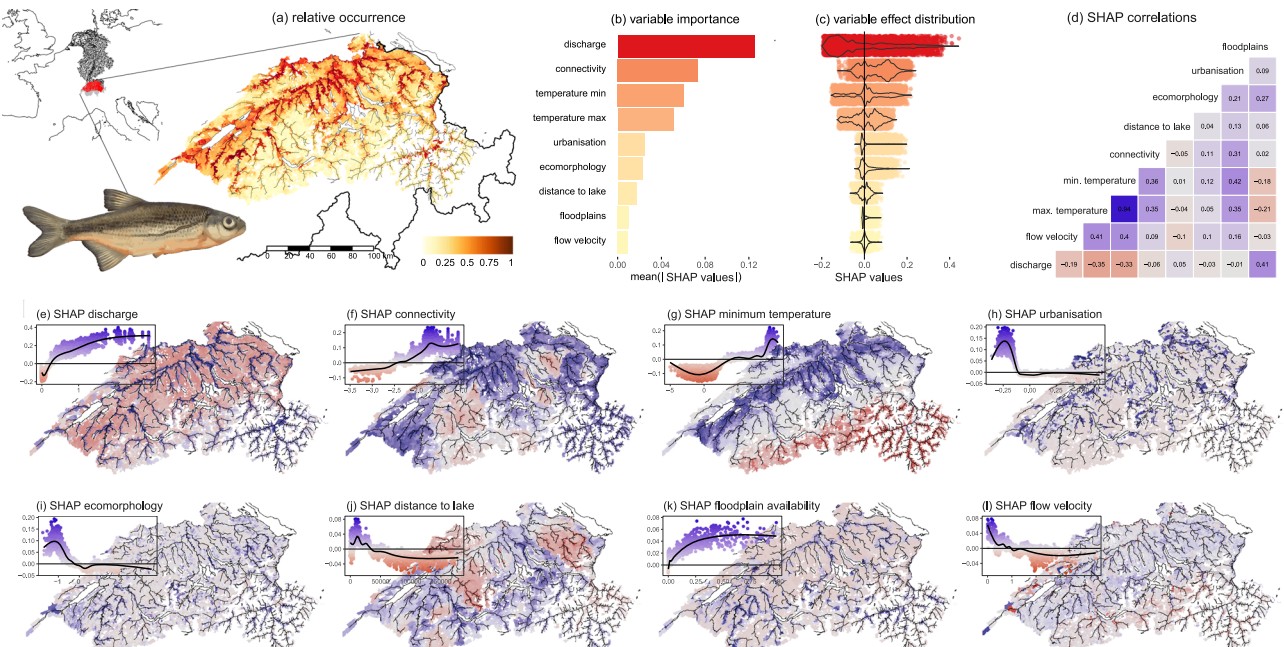

**Fig. 2 | Decoupled spatial drivers of species occurrence for *Alburnoides bipunctatus* in the Aare-Rhine catchment of Switzerland. a** shows the predicted occurrence from a random forest model fitted to presence-absence data, the map inset places our focal catchment region (red) in the wider Rhine River watershed (black). **b** shows variable importance calculated as the average of absolute SHAP values in the calibration dataset, indicating the average variable contribution to the overall prediction in the calibration dataset. **c** shows the distribution of SHAP values per variable, indicating whether contributions are positive or negative. Red intensity indicates variable importance in (b–c). **d** shows the pairwise Spearman's rank correlation between each variable's SHAP values, indicating the extent to which variable contributions to predictions are spatially correlated. Blue indicates positive correlations, and red indicates negative correlations. **e–l** shows the spatial pattern of SHAP values and, therefore, indicates the contribution of a variable to a prediction in a given sub-catchment. We do not show the maximum temperature to simplify the figure because it was a highly similar pattern to the minimum temperature. Positive (blue) and negative (red) contributions to the environmental suitability prediction show distinct spatial patterns to each threat, as also indicated in (**d**). Insets show the environmental values on the *x*-axis and SHAP values on the *y*-axis and, therefore, indicate the shape of the species response curve (see Supplementary Fig. 18 for response curves across all species). Note the change in scales between panels indicated by the *y*-axis of the insets. Data required to reproduce this Figure is available in Supplementary Data 1 of our Figshare repository[86].

catchments had positive contributions for every individual natural abiotic factor.

Next, we investigated anthropogenic threats within the expected distribution, finding that in 89% of *A. bipunctatus's* expected distribution, there was at least one threat with a negative contribution to environmental suitability (one threat in 29% of areas, two in 33%, three in 22% and four in 5%). Within the expected distribution, we summed all potential positive and negative threat effects and found a net negative effect of threats in 14% of sub-catchments. In sub-catchments with at least one threat, we also found a 27% reduction in environmental suitability compared to unthreatened sub-catchments. Consequently, within the expected distribution, the rate of predicted absence was 1.66 times higher in threatened (78% absent) compared to unthreatened (47% absent) sub-catchments (Fig. 4c, d).

Quantifying which threats act within the expected distribution revealed counter-intuitive impacts of threats on species' distributions (Fig. 5a, b). Specifically, (lack of) connectivity was often the most important threat variable in our model (Figs. 2b, 3). However, much of the negative impact is outside of the natural abiotic realised niche of the species. As such, only 15% of areas inside the expected distribution for *A. bipunctatus* were negatively impacted by a lack of connectivity – i.e., areas falling inside the abiotic niche were generally well connected (Fig. 4c). We found the opposite for habitat quality indicators which had lower overall model importance, but had negative contributions to environmental suitability across a larger percentage of the expected distribution (e.g., high river morphological modification index = 47%, low floodplain cover = 62% and high urbanisation = 57%). These single-species results were highly consistent when assessed across all species

independently, with an average of 88 ± 9% sub-catchments inside species' niche having a negative contribution of at least one threat, and 23 ± 12% sub-catchments having a net negative effect of all threats (Fig. 5a–c, Supplementary Fig. 19 and Supplementary Table 4).

We investigated multi-species quantitative shadow distributions by calculating the percentage reduction in suitability predictions when including threats compared to when excluding threats (i.e., the expected suitability; Fig. 5a–c). Across all sub-catchments, we found environmental suitability in the observed distribution was reduced by 24% (averaged across species per sub-catchment = 0.38) compared to the expected distribution (0.54; Fig. 6a compared with 6b; two-sided t-test, t = −100; *p* < 0.001). This average shadow distribution was spatially heterogeneous (Fig. 6d). Some large contiguous patches had lower than expected suitability, with 10% of areas having suitability reduced to at least 56% of the expected suitability (Fig. 6c, d). Hiding beneath these across-species averages were also strong reductions in suitability for certain species within the expected assemblage (Fig. 6e). As such, the most negatively impacted species in each sub-catchment had environmental suitability reduced to only 58% of the expected suitability on average across sub-catchments (Fig. 6e). The mean shadow distributions correlated negatively with both habitat quality and connectivity (Fig. 6g, h; Spearman's rank correlation, $\rho = -0.69$; *p* < 0.001; $\rho = -0.63$; *p* < 0.001) but less for minimum shadow distributions for habitat ($\rho = -0.49$; *p* < 0.001) than connectivity ($\rho = -0.68$; *p* < 0.001), indicating connectivity constraints defined the most negative impact within a community. We found the general spatial patterns of shadow distributions were highly consistent regardless of how shadow distributions were estimated but, as

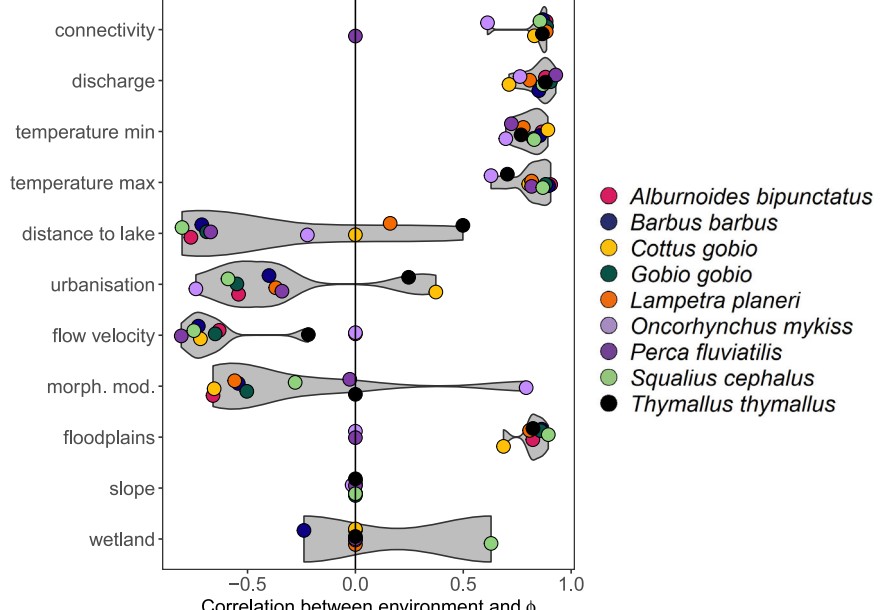

**Fig. 3 | Variation in environmental effects on species distributions amongst species and between environmental factors.** The *x*-axis shows Spearman's rank correlation between environmental values and SHAP values, with high positive or negative values indicating coupling and a strong contribution of the variable to determine species spatial variation in environmental suitability. The variation in Spearman's rank values between species indicates the magnitude of variation (or lack of) in species responses to a given variable. Data required to reproduce this Figure is available in Supplementary Data 2 of our Figshare repository[86].

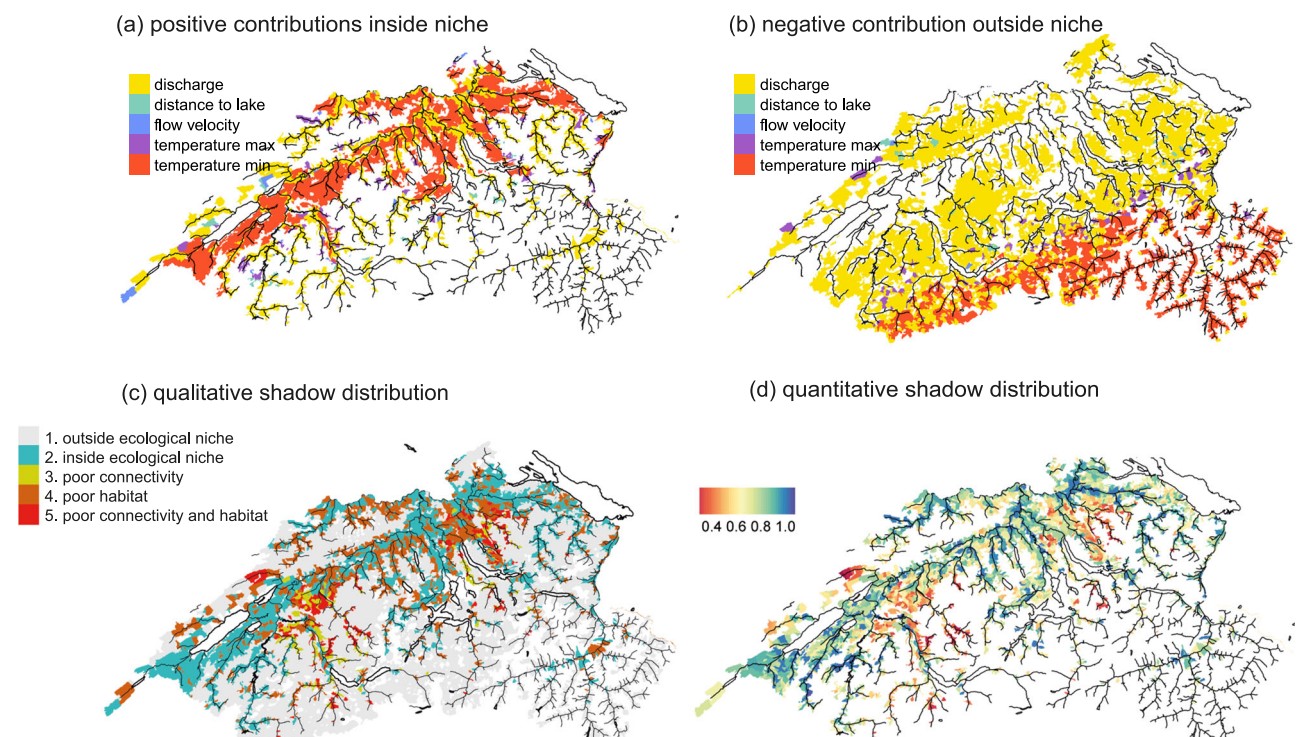

**Fig. 4 | Evaluating the relative contribution of natural factors and anthropogenic threats to species distributions using SHAP values for *Alburnoides bipunctatus*. a**, **b** SHAP values per sub-catchment for abiotic niche factors that sum to be positive (**a**) or negative (**b**). Colours represent the largest contributing abiotic niche factor (discharge, flow velocity, minimum and maximum temperature and distance to the lake). **c** The qualitative shadow distribution for *A. bipunctatus* indicating sub-catchments within the abiotic niche but having negative connectivity SHAP values (poor connectivity), negative habitat quality SHAP values (poor habitat) or negative habitat quality and negative connectivity SHAP values (poor connectivity and habitat), as well as no negative SHAP values for threats (blue). Grey indicates areas outside the ecological niche (as in **b**). **d** The quantitative shadow distribution for *A. bipunctatus* shows the ratio between observed suitability and expected suitability, with red indicating a stronger shadow distribution (lower than expected suitability) and blue indicating environmental suitability scores close to expected suitability. Data required to reproduce this Figure is available in Supplementary Data 1 of our Figshare repository[86].

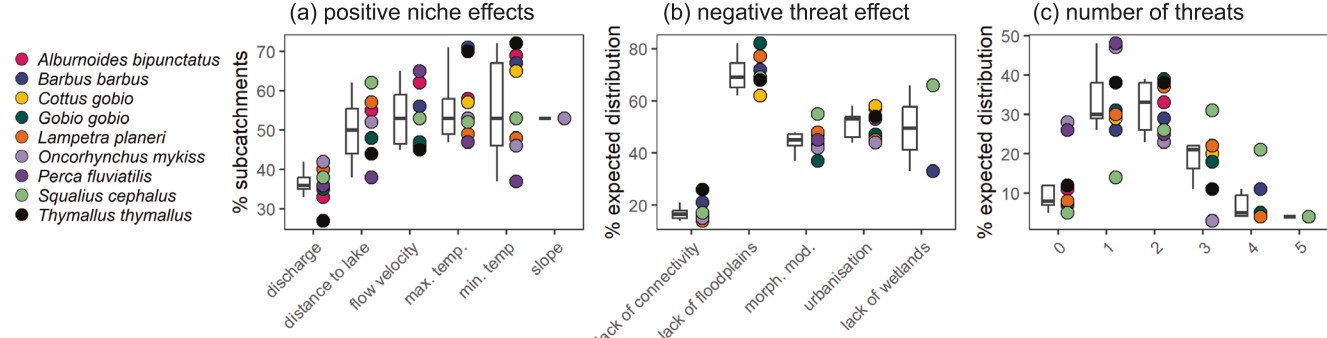

**Fig. 5 | Deconstruction of geographic ranges reveals counterintuitive threat impacts within expected distributions. a** The percentage of sub-catchments with positive abiotic niche-related SHAP values. A high proportion of available sub-catchments have positive effects of abiotic niche variables, which together define the expected distributions of species, but discharge contributes positively in the fewest suitable sub-catchments. **b** The percentage of sub-catchments inside the expected distribution with negative threat SHAP values. Lack of connectivity, despite being the most important predictive threat, has a negative impact in the fewest sub-catchments within the expected distribution of most species. **c** The percentage of sub-catchments with a specified number of negative SHAP values for threat factors. Boxplots indicate the median (solid line), inter-quartile range (hinges) and 1.5*IQR (whiskers) for properties of the 9 studied species. A large proportion of all species expected distributions have one or more threats acting negatively on environmental suitability scores. Data required to reproduce this Figure is available in Supplementary Data 3 of our Figshare repository[86].

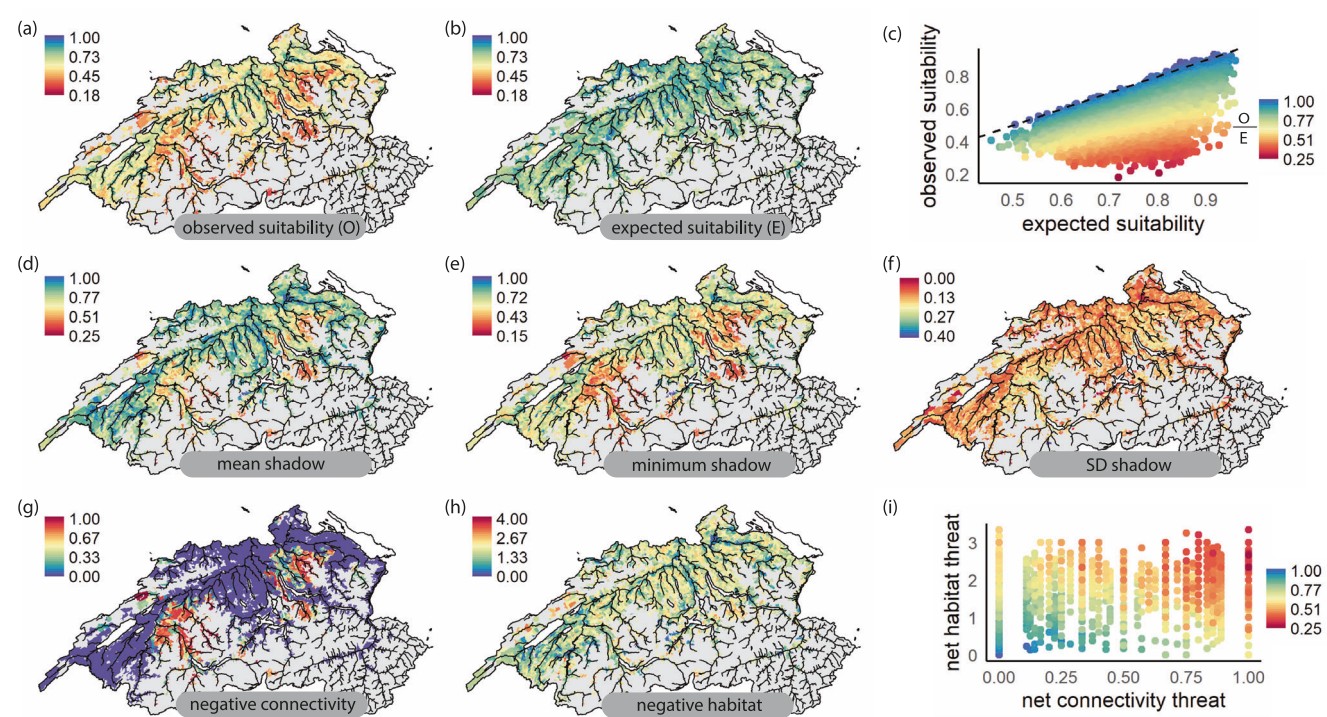

**Fig. 6 | Multi-species average shadow distributions and their constituent parts per sub-catchment.** Panels show average (**a**) observed suitability which includes the influence of threats, (**b**) expected suitability as the suitability excluding the influence of threats, (**c**) biplot of average expected vs. observed environmental suitability (i.e., predicted from models without SHAP adjustment) values coloured by the mean. The summary of shadow distributions was calculated as the ratio between observed and expected suitability (maximum of 1 and minimum of 0) and is shown as an average across species (**d**), as the minimum value for any species in a sub-catchment (**e**) and as the standard deviation amongst species within a sub-catchment (**f**). Shadow distributions are contributed by threats that negatively affect species distributions (**g**, **h**), represented here as the occurrence (0 or 1) of negative contributions for connectivity (**g**) or habitat quality (**h**, floodplains, wetlands, urbanisation, river morphological modification index) summed within a catchment and averaged across species. For example, where three of four habitat quality indicators negatively impact a species in a sub-catchment, the catchment receives a score of three. **i** shows a biplot where points are sub-catchments coloured by the mean shadow distribution value for that sub-catchment. Note that to avoid uncertainty in averages containing few species, only sub-catchments with more than 2 species in the expected distributions are shown, areas with 2 or fewer species are therefore indicated in grey. Data required to reproduce this Figure is available in Supplementary Data 3 of our Figshare repository[86].

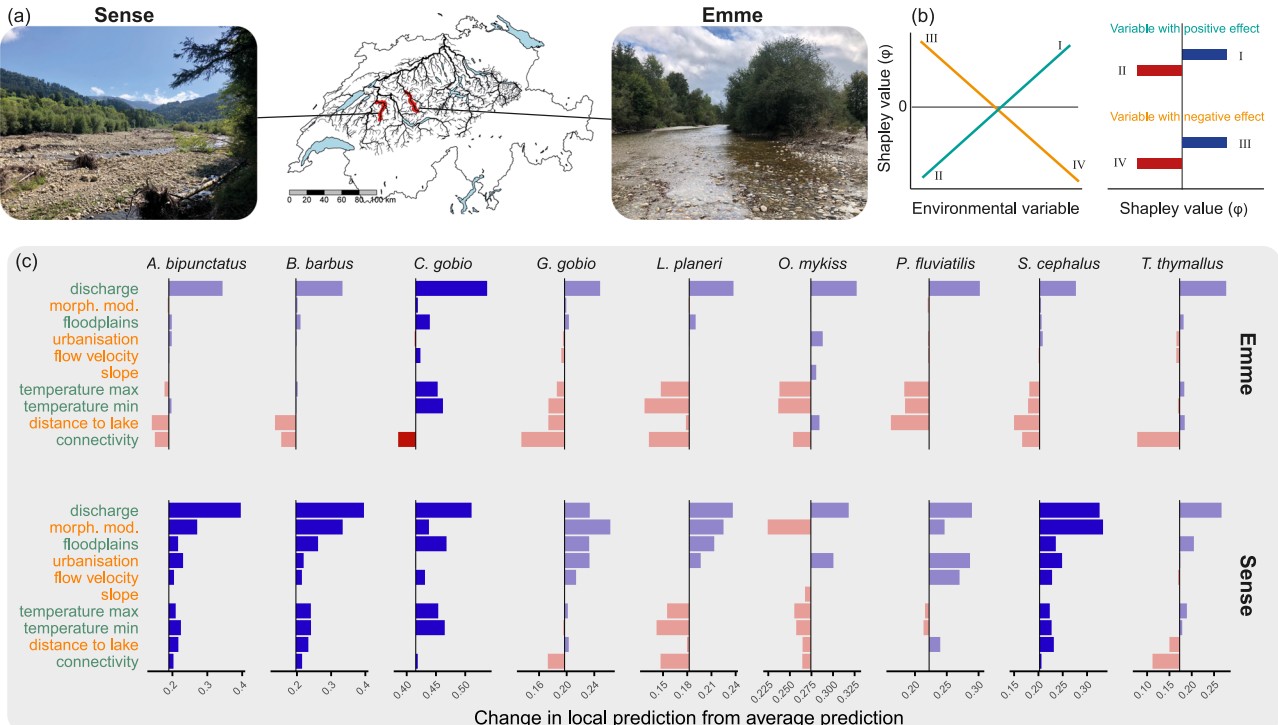

**Fig. 7 | Relative local contribution of variables to environmental suitability predictions, as indicated by SHAP values, showing species local sensitivities to each environmental factor in the upper Emme and Sense River catchments for 9 species. a** shows the location and habitat of the Sense and Emme Rivers in the Aare catchment. **b** shows the contrasting response direction, positive or negative, and the interpretation of a positive or negative SHAP value for variables with different effect directions. **c** shows the local variable contributions expressed as the deviation from the mean occurrence prediction for each species in the model (vertical black lines). In (**b**), when variables have a positive effect (green, I, II), a positive SHAP value indicates that the high values of that variable contribute positively to environmental suitability, as shown in I (or vice versa in II). Where

variables have a negative effect (orange, III, IV), a positive SHAP value indicates that the high values of the variable negatively contribute to environmental suitability. Variable colours in (**c**) indicate whether the effects are positive or negative, as in (**b**). Bar colours indicate the sign of the SHAP value of effect (red = negative, blue = positive). If species were recorded in the catchments in surveys, they are indicated in dark shading, in contrast to light shading where species were absent from available records. The explanatory variables used in the model are ordered by average SHAP values across both catchments, from positive to negative. Data required to reproduce this Figure is available in Supplementary Data 2 of our Fig-share repository[86].

expected, exhibited reduced loss of environmental suitability (Supplementary Figs. 6, 20, 21).

### Local relative contributions between contrasting locations

We contrast the more modified Emme River to the more natural Sense River to reveal the localised impacts of different environmental factors on species in different management contexts. We found the low connectivity in the upper Emme River contributed negatively to most species' environmental suitability (i.e., negative SHAP values), and many species were absent from this catchment (Fig. 7). The species found in local surveys (*Cottus gobio*) had weak overall contributions of connectivity to environmental suitability (SHAP values near zero; Fig. 7). The high distance from lakes in the upper Emme River also contributed negatively to the environmental suitability for *Barbus barbus*, *Perca fluviatilis*, and *Squalius cephalus* even though the flow velocity in the Emme contributed positively to the environmental suitability for these species. In the Emme River, morphological modification index, urbanisation, and floodplain availability had SHAP values near zero for most species. In contrast, the Sense River had strong positive contributions of variables related to habitat quality across multiple species, which were also directly observed in local surveys. These positive contributions of high habitat quality to environmental suitability were counteracted by the negative contributions of connectivity for some species (e.g., *Gobio gobio*, *Lampetra planeri*, *Thymallus thymallus*), leading to lower-than-expected species environmental suitability predictions for these species.

## Discussion

Our framework attributes variation in population performance between specific locations to the environmental conditions and potential threats at those locations. Broadly, our approach using model agnostic explainable AI tools is generalisable to any ecological or evolutionary property, such as species occurrence, abundance, reproductive rate and genetic diversity for the purpose of quantifying species-specific responses of populations to environmental gradients[33]. We provide two main contributions, which further the understanding of species' realised spatial distributions and, in turn, aid biodiversity conservation efforts. First, we partitioned the spatial drivers of species distributions into natural and anthropogenic factors, introducing the concept coined a '*shadow distribution*' (Fig. 1). This concept provides valuable insights into hidden environmental and human impacts on species' geographic range. Second, we address a long-standing ecological challenge and offer a solution to quantify the spatial manifestation of species-specific responses to multiple environmental gradients[23,24]. This contribution enables the relative importance of local conditions for species communities to be understood, supporting conservation decision-making (Fig. 7).

### Shadow distributions and spatial drivers of geographic distributions

To our knowledge, our study is the first to define and quantify a property like the '*shadow distribution*'. Combining insights from XAI with ecological principles allowed us to partition the available range for a species into two parts: the area falling inside the ecological

niche where a species could persist, and the area inside the ecological niche but where humans negatively impact species' populations. Our solution, therefore, reveals the spatial variation in areas negatively impacting species' populations and locations where natural factors support species. This addresses a critical gap in ecological niche theory: that human influences should be better understood as fundamental determinants of species' geographic distributions[10]. Our results imply that if omitting the negative impact of anthropogenic threats on environmental suitability scores from SDMs, i.e., ignoring shadow distributions, then biodiversity mapping is likely to underestimate the potential distributions of species. As such, opportunities for restoration may be overlooked because species are assumed to be naturally absent when species are actually absent due to threats. Shadow distributions could, therefore, assist researchers and practitioners in moving beyond mapping threats or species distributions alone. Shadow distributions could also help to better define reference ecological conditions based on the expected distribution of species, given their natural ecological niche. We note that previous work has separated out the abiotic drivers of species distributions for African elephants[23] (*Loxodonta africana*) or demonstrated how relationships between environment and performance can vary spatially (e.g., non-stationarity)[34,35]. However, these studies do not identify the relative contributions of natural vs anthropogenic threats in a spatially explicit way that allows the definition of shadow and expected distributions.

Revealing the shadow distribution of species opens many new questions that we can only partially answer here. For example, what factors, both intrinsic and extrinsic to species, determine how and why species differ in their shadow distributions? Our species comprised widespread temperate freshwater fishes with shadow distributions that were quite consistent. This consistency likely reflects shared negative responses of species to a lack of river connectivity and to indicators of river habitat quality (e.g., Fig. 3). Other ecological systems and species groups could have varying patterns, depending on: (i) the number of threats, (ii) the strength of a threat's effect, (iii) the diversity of responses in a community to a threat, (iv) the abiotic niche breadth for natural factors or ecological versatility of species and (v) the diversity of niches in the community. Shadow distributions are likely to be larger in more sensitive species facing multiple significant threats and having narrower abiotic niches. Shadow distributions may be most consistent among species when the community response diversity is low, and species share abiotic niche preferences. Future research should discern how each of these factors independently modifies the size and structure of species' shadow distributions and variation among species. A research agenda on shadow distributions, i.e., areas where threats negatively impact species natural distributions, would help identify conservation areas beyond current biodiversity hotspots or areas of potential threats and instead focus on areas where evidence indicates negative responses to threats given high expected suitability and diversity. In addition, the reconstruction of expected and reference community states could occur through summarising the shadow distributions across multiple species.

Fundamental questions on how species geographic ranges are spatially structured can be asked and answered with our framework. For example, the relative role of abiotic or biotic drivers at equatorward and poleward range edges has long remained elusive[36,37]. Our results also indicate that environmental determinants of range boundaries and internal range structure can differ. Exploring how internal range structure is environmentally determined is an emerging field. In general, the factors influencing the internal structure of a geographical range are expected to differ from those affecting geographical range boundaries[38]. Recent research has revealed that climate change can modify internal range structure independently from range edges, which may arise if metapopulation viability differs between range interior and range edges[39]. For *Alburnoides bipunctatus* and several other species, minimum temperature emerged as a

primary negative factor at the cold-alpine range limit despite many rivers having adequate discharge. Conversely, within the accessible thermal niche, we found patchy distribution driven by insufficient discharge or low habitat quality.

The shadow distributions revealed here confirm the detrimental effect of hydromorphological alterations on river fish populations, which more widely drives the poor quality of European rivers for biodiversity[25,40,41]. The uncertainty in biodiversity and threat data makes it difficult to causally attribute biodiversity change to drivers[42]. The mechanistic knowledge of the causes of population declines rarely exists (but see ref. [43]), so it is often still necessary to speculate on the causes of biodiversity change in each specific context, relying on expert knowledge or anecdotal inference[44]. Even though our study focused only on a coarse resolution 'presence-absence' biological response we recovered associations indicating impacts from multiple co-occurring threats. Our findings suggest a milieu of threats together reduce population performance across the riverscape, leading to absences from multiple locations with negative impacts indicated by: (i) low longitudinal connectivity due to high river barrier density; (ii) low physical complexity of rivers; (iii) a lack of natural spatiotemporal fluvial dynamics that generate riparian floodplains and instream habitat variation and (iv) distance to urban areas. Which threat acts where and how is then revealed through our XAI approach. Our findings broadly support findings from more intensive single-species studies that show reductions in recruitment, growth rates, survival and migration success from similar threats[43,45].

## Species-specific responses to multiple environmental gradients

We illuminate how unexpected outcomes of ecological management can arise: each location in our analysis had environmental factors with high importance and low importance for each different species, which also shifted between locations. Equipped with this knowledge, environmental managers can identify the factors that support or impede population performance for each species in specific locations. In turn, this enables more accurate expectations of local-scale biodiversity responses to restoration and management. In addition, knowing if species have divergent or similar environmental responses indicates whether actions support whole communities or individual species (e.g., Fig. 3). Furthermore, we provide insights for spatial conservation planners to account for the spatial sensitivity of species to threats in order to facilitate planning of multiple conservation actions across a land- or riverscape[46]. A lack of experimental evidence on population constraints in specific locations for various species hinders managers from accurately evaluating the impact of a single threat amid multiple factors. In river systems, neglecting to address multiple threats simultaneously prevents biodiversity recovery. For example, under habitat restoration, critical threats such as connectivity are often ignored while habitat quality is addressed but only weakly limits populations[12,47]. Our work helps address the challenge of assessing the potential success of different conservation options, which is crucial for improving long-term management outcomes for biodiversity[48].

Whilst the diversity of species responses to environmental gradients is well-recognised, our work can help reveal how observed local biodiversity, and biodiversity change, arise from independent responses of different species to environmental change in any specific location (e.g., Fig. 7). Previous work often veils the complexity of species responses to environmental gradients behind an overall prediction of occurrence (or abundance) in a given location, or a measure of community stability[49]. This is almost always the case with predictive models of species distributions or abundance[20,50] (but see ref. [23]). Combining AI tools with phenomenological models of natural systems can accelerate valuable insights on how multiple threats impact biodiversity - insights traditionally only identified through expensive multi-factorial experimentation[51].

## Cautions, limitations and future work

When quantifying species' shadow distributions we suggest a cautionary inferential approach where each species' response curve is well understood and trusted before application of our framework for shadow and expected distributions. This potentially limits the number of species studied and the spatial scope of analyses. This guidance contrasts with many applications of species distribution modelling and machine learning[50] where ecological phenomena are predicted with relatively high accuracy but not necessarily well understood, and then applied at global scales[52] or to tens of thousands of species[53]. However, making decisions often requires well-understood models in more local-to-regional contexts to ensure better matches between information needs and decision contexts[54]. All applications of decision-making contain costs, such that trusting predictions of black-box models risk inefficiencies and wasted resources[23,55]. In many such cases, interpretability and explainability could be a higher priority than the traditional focus on predictive accuracy and empirical cross-validation by site-specific prediction tests is recommended.

We attempted to ensure that each variable could be partitioned as an independent effect on environmental suitability, however, even without multicollinearity issues, the use of XAI does not necessarily imply causality[56,57]. It should be noted that SHAP values are not inherently robust to correlated features, and we needed to check the multi-collinearity of variables, decorrelate variables that exhibit (non-linear) dependence, and remove variables that exhibited biologically implausible relationships (see ref. 30 for further discussion). Our need to use de-correlated variables echoes deep issues underlying the philosophical foundation of statistical modelling of observational data that are still imperfectly addressed across multiple scientific domains and could be further improved[56,58]. Users should also recognise that SHAP values do not enable actions directly[59] without first understanding the direction of species response curves to threats and where locations fall on this curve which enables contrasts with other observations. For example, in Fig. 7, the directionality of SHAP values only makes sense in considering the overall response direction (Fig. 7b). We caution that in estimating shadow distributions, the choice of adjustment to the "reference" state should be carefully explored. Furthermore, we note that other XAI tools, such as counterfactual explanations, could provide additional insights into the potential impact of alleviating threats or further exploring scenario building by modifying the feature space, which is common practice in biodiversity projections[60].

We note that XAI-based model interpretations can only be as good as the quality of the model performance (which here ranged from AUC of 0.66–0.93), and an accurately performing model must represent realistic biological phenomena. These points ultimately depend on the quality of the biological and environmental data input into the model, secondarily on modelling steps such as the choice of SDM algorithm and XAI method. For example, in our study, we could not yet include rarely-available local variables such as multiple forms of pollution, the alteration of natural flow regimes through hydropower generation, and the location of extreme drought or thermal events[61]. In addition, our natural abiotic variables represent gradients along the natural river continuum (cold, fast flowing, small headwater streams to warm, slow flowing, large main stems) rather than human impacts on river temperature, flow and discharge regimes[29]. Including finer-scale anthropogenic variables may recover additional important conservation-related responses to threats that are currently missing from our shadow distribution estimates. We opted for the simplicity of a single algorithm (random forests), but ensembles of SDMs applied to SHAP could help reveal sources of uncertainty in shadow distributions[62]. This approach would require careful validation of all response curves in all model types put in the ensemble. Further, we chose SHAP as our XAI tool, which is a source of unexplored uncertainty, but many other tools

exist with different mathematical axioms, some of which may provide alternative insights (see ref. 33,63,64).

Some more conceptual caveats applicable to all SDM models also apply to the interpretation of shadow distributions, for example, whether the model of the realised niche accurately represents the species fundamental niches influences how well the deconstructed environmental contributions reveal the shadow and expected distributions. Future work could also better reveal how species interactions influence species environmental suitability and shape expected distributions (e.g., reduced expected distributions through competitive exclusion)[65]. Further, any issues relevant to presence-only SDMs, such as sampling biases, are also problematic for SHAP explanations of these models, and as such, we encourage the use of presence-absence data from standardised surveys to build SDMs. We also note that probabilistic presence-absence surveys can still contain biases because some sampling methodologies bias against difficult to sample habitats, here large rivers, which biases the amount of data available across habitat gradients. The limitation that environmental suitability predictions should relate to population performance for valid biological interpretation[66] also applies here.

Revealing whether shadow distributions correlate with declines in genetic diversity, demographic rates, and population abundance is an important future validation. Further validations through field manipulations of threats could empirically test whether conservation gains are greater when guided by the outputs of large-scale modelling exercises. Future work could also better attribute range loss to spatial threats using more direct measures of population performance, such as local abundance, age structure or population health, especially at the edges of species' ranges. Because environmental suitability often non-linearly relates to population viability and potential ecosystem service provision, we likely underestimate ecological consequences of threat impacts and, therefore, the size of shadow distributions, using presence-absences[50].

In conclusion, for biodiversity conservation, protection and recovery, we must identify and contextualise threat impacts within the multiple natural constraints on species distributions. We show how to identify when threat impacts occur in portions of species' geographic distributions that are naturally highly suitable. We highlight an important decoupling between the different factors that determine species distributions. We define species' expected distribution and species' shadow distribution to help quantify the magnitude of this decoupling. Our work suggests indicators for national Biodiversity Action Plans underlying the Kunming-Montreal GBF based on species distribution models should also consider expected and shadow distributions. Failing to do so, we miss insights to the negative influence of anthropogenic threats on species distributions. Our work supports the assessment of threats to biodiversity at large scales and moves towards a framework tailoring conservation actions to local threats demonstrated to impact species distributions.

## Methods

Our research complies with all ethical regulations being collected under the Swiss animal experimentation licences issued by the Kanton Bern (Office of Veterinary Affairs; permit numbers 34546 BE11/2022 and 34150 BE95/2021).

### Overview

We used a species distribution modelling approach to model the environmental suitability across the spatial distribution of nine fish species in Switzerland using 11 environmental variables. We next applied model agnostic explainable artificial intelligence tools to these models. These tools calculate the local relative contributions of each environmental variable to the prediction of environmental suitability at the observation level (here, 2 km sub-catchments). To focus on the main aim of our work – to investigate the local relative contribution of

each variable for each species in each location – here we provide only an overview of the underlying species distribution model protocol and provide a full 'ODMAP' protocol in the Supplementary Methods[67].

## Species data

We focus on rivers and streams in the Aare, Limmat, Reuss and Rhine catchments within the political boundaries of Switzerland. These river catchments drain the northern slopes of the European Alps and together drain an area of 28,057 km$^2$ into the main Rhine catchment. The native fish fauna share a common biogeographic history. We focus on nine example species: *Alburnoides bipunctatus, Barbus barbus, Cottus gobio, Gobio gobio, Lampetra planeri, Oncorhynchus mykiss, Perca fluviatilis, Squalius cephalus, Thymallus thymallus*. Note that *Cottus gobio* is likely a species complex[68]. We selected this set of species to cover a wide range of ecological preferences with some species being nationally threatened with uncertain drivers of population declines and range loss (OFEV / CSCF 2022; note that *O. mykiss* is a non-native species that we include for contrast), while others are common and important components of river ecosystems.

We compiled quantitative electrofishing surveys that provide presence-absence records. Field surveys were conducted by scientists, governmental monitoring and environmental consultancies (see Supplementary Table 1 for an overview of species by survey data and ODMAP protocol). Fish richness and composition have been shown to vary little between electrofishing fishing crews or methods such that our data synthesis is assumed to be robust against potential systematic biases introduced by combining datasets[69]. We performed our analyses on data collected after 2010 to avoid potentially including records that indicate species presence before the modern threats have impacted species' populations causing local extirpation. Supporting analyses confirmed that, in general, there was higher performance for models combining all available data (Supplementary Fig. 1), which together provided a more complete coverage of environmental space (Supplementary Fig. 2). In total, we analysed 38,100 records of species presence-absence containing 1933 presence records for 3216 sites surveyed between 2010 and 2023. There were, on average, 180 presence records per species ranging from 48 (*Thymallus thymallus*) to 791 (*Cottus gobio*).

## Environmental data

We compiled and processed data on the spatial distribution of 18 environmental variables representing a range of natural and anthropogenic threat variables to use as covariates in our models (see ODMAP Protocol). We attempted to cover a wide range of in-water and riverscape environmental gradients by compiling the following variables [short-hand name] for consideration in our models: maximum annual discharge [discharge], minimum slope [slope], mean flow velocity [velocity], mean annual temperature [mean temperature], maximum annual temperature [maximum temperature], minimum annual temperature [minimum temperature], colonisation probability index [connectivity], minimum distance to the lake [distance to the lake], river morphological modification index [morphological modification], proportion cropland cover [cropland], mean tree cover density [tree cover], mean surface imperviousness [urbanisation], mean livestock unit density [livestock], proportion wetland habitat [wetland], proportion floodplain habitat [floodplain], mean diffuse nitrogen inputs [nitrogen], mean diffuse phosphorous inputs [phosphorous], and mean insecticide application rates [insecticide]. We qualitatively evaluated the expected spatial scale of effect on freshwater fish species distribution based on review, elicitation, and discussion amongst co-authors and processed data according to the greatest expected scale of effect (see ODMAP protocol). Depending on the variable and the dataset, we calculated variables at three potential spatial scales, (i) the local values within 100 m from the river, (ii) sub-catchments characterising lateral overland flow, (iii) upstream catchment representing accumulation of environments over a larger spatial scale than reach contributing areas (see ODMAP for full details). For convenience during later analysis steps, environmental data were harmonised to a common 100 m by 100 m raster grid with an equal area projection for Europe (ETRS89-extended/LAEA Europe), and model predictions were aggregated to river sub-catchments using the 'Topographical catchment areas of Swiss water bodies 2 km$^2$' Federal Office for the Environment data product.

## Model fitting and evaluation

We fitted 'down-sampled' random forests (sensu[70]) for each species using environmental data as covariates and species' presence (1) or absence (0) as response variables. Random forests perform well at prediction tasks across multiple data types and have been demonstrated to perform as well as model ensembles in modelling species distributions[50,71]. A major benefit of random forests is the automatic recovery of non-linearities and variable interactions. We used down-sampling to address the class imbalances that can lead to models overfitting training data if absences far outweigh presences[70]. In this down-sampling procedure, each tree is fitted to a balanced sub-sample of presences and absences. As such, model predictions are not strictly probability of occurrence because presences and absences are balanced and, therefore, instead represent and index of relative occurrence. We refer to predictions as 'environmental suitability' for consistency with SDM literature, although this term is often used for predictions of presence-only models. We set the ntree parameter to 1000, the downsampled 'sample size' to be the minimum of either class (0 or 1), and set the mtry parameter to the square root of the number of covariates. We follow[70] in not further tuning random forests parameters which exhibit low tuneability[72,73]. Random forests were fit using the R package randomForest (version 4.7–1.1[74])

A fundamental aim of our work is to provide interpretable (understanding inner workings) and explainable (understanding why a prediction is made) models. Multi-collinearity induces challenges in interpreting the independence of variable effects and interpretation of SHAP values[59]. Through the below procedure our final variables were highly decoupled having a median absolute correlation of 0.05, a 95th quantile of 0.26 (Supplementary Fig. 3). We therefore limit the impact of multicollinearity in our modelling (see Supplementary Methods for full details). We first checked bi-plots and Spearman's rank correlations between variables and identify potentially confounding factors that would lead to misinterpretation of focal variable effects. We found elevation, discharge, slope and distance to lakes were often strongly related to 8 variables (morphological modification, urbanisation, livestock, nitrogen, phosphorous, insecticide, cropland, and tree cover). We then fitted GAMs to relate these variables with the potential confounders and used the residuals from GAMs in our random forests. We retained only residual morphological modification and residual urbanisation, which had biologically realistic relations with environmental suitability. The interpretation of these processed variables is the relative value of the variable given the site's elevation, discharge, slope and distance to the lake. GAMs were fitted using the R package 'mgcv' (version 1.8–38)[75]. From our final pre-selected set of variables, we then identified and used only those that were statistically supported using the BORUTA algorithm in the R package 'BORUTA' (version 7.0.0)[76]. This method was developed to provide a statistically valid approach to remove variables that do not sufficiently improve the fit of random forest models[76].

We first assessed model performance using spatially blocked 5-fold cross-validation by iteratively fitting models to training sets (4/5 folds) and predicting occurrence in testing sets (1/5 folds). We used the R package 'blockCV' (version 3.1–4) and set the distance to 10 km, which in preliminary assessments emerged as the scale of environmental autocorrelation in our covariate data[77,78]. We evaluated model performance using 13 metrics of model performance (see ODMAP

protocol), but focused on the True Skill Statistic (TSS), Matthew's Correlation Coefficient (MCC) and area under the receiver operating characteristic curve (AUC) as integrative measures of performance across the contingency matrix (Supplementary Fig. 4 and Supplementary Tables 2, 3). Random forests presented in the main text were then fitted to all available data for each species.

## SHAP values: estimating local relative contributions of variables to species environmental suitability

We aimed to quantify the effect of an individual covariate on a species occurrence in a particular location, sometimes referred to as the 'local feature importance', the 'situational importance' or 'local contribution' of a variable (Fig. 1). To do so, we approximate Shapley values defined in game coalition theory[79], which are called SHapley Additive exPlanations (SHAP) when applied to explain machine learning predictions[30,59,79].

SHAP values are an explainable artificial intelligence (XAI) tool to explain a prediction made by a model. XAI methods aim to explain why complex "black box" models made predictions at an observation level. SHAP values are one tool to provide an interpretation of the covariate effect on the predicted outcome at the observation-level in the model. A SHAP value indicates the difference between what a variable contributes to a prediction in each location, and what the variable is expected to contribute given the mean model prediction. Other variable importance approaches generally provide 'global' insight to variables importance across all observations in the model (e.g., permutational variable importance). In contrast, SHAP provides a single value per observation per variable. This SHAP value indicates the features contribution to the prediction for that specific data point. In a spatial model, the observation level is inherently linked to locations. In our models of species occurrence, a positive SHAP value indicates a given variable is contributing positively to the environmental suitability prediction (increases the prediction), and vice versa, and if it is 0, it has no contribution. We can compare SHAP values of all other variables in the focal location to understand the relative importance of individual variables within a species distribution. Or, for the same site we can compare between species the relative contribution of different variables. SHAP values are model agnostic and so can generalise to any statistical model that explains variation in ecological properties across environmental gradients (e.g., abundance, biomass, growth rates, body condition, productivity, species richness). The application of XAI in ecology and conservation is nascent[22,23,55,62,80-83] and so we provide a detailed explanation of SHAP values in Supplementary Note 1.

In addition to local interpretations, aggregating SHAP values across all observations in a model gives an indication of 'global' variable importance[63]. Due to SHAP values satisfying the efficiency criteria of interpretable XAI methods[30] (summing to the predicted mean), summing subsets of variables by groups indicates contributions of groups of variables to the mean prediction (e.g., summing SHAP values across all threat variables). We calculated the mean absolute SHAP value, which indicates a variable's overall importance in changing model predictions. We also correlated SHAP values against original environmental values, which indicates overall response curves between variables and model predictions. Note that the overall importance of variables in determining species range-wide distributions was comparable to traditional measures of 'global' variable importance, such as permutational variable importance scores (Supplementary Fig. 5).

We used the Štrumbelj & Kononenko[85] Monte-Carlo approach using 10,000 repetitions to calculate SHAP values from the down-sampled Random Forest model, using the *explain* function in the R package '*fastshap*' (version 0.1.1)[84]. Using SHAP values to calculate observation-level variable contributions has benefits over other interpretable machine learning approaches, such as LIME, breakdown, or counterfactual explanations, by satisfying the efficiency, symmetry,

dummy and additivity properties[79,85] (see ref. 30,33,59 for further discussion). Note, however, that multiple options exist for calculating local model explanations and model-specific faster alternatives for tree-based methods exist that are a better alternative for larger datasets with more features, such as "*TreeExplainer*"[63].

## Addressing ecological and conservation challenges with local relative contribution of variables

**(i) Mapping local relative contributions for single- and multi-species comparisons.** Here, we provide an in-depth exploration of local relative contributions of variables, as quantified using SHAP values, for the geographic distribution of a single species, the spirlin, *Alburnoides bipunctatus*. We chose *A. bipunctatus* because it is a relatively widespread species in our catchments and is classified as 'Vulnerable' based on apparent population reduction (criteria A2c) and extent of occurrence < 20,000 km² with a continued decline in the area and quality of habitat (criteria B1biii). We summarised the SHAP values of each environmental factor to *A. bipunctatus* environmental suitability prediction in each river sub-catchment. We mapped SHAP values for each variable to explore the spatial distributions of variable contributions. We performed pairwise Spearman's rank correlations between all variables to assess whether SHAP values for different variables within one species had different spatial distributions. We also calculated the global variable importance as the mean absolute SHAP value per variable across all sub-catchments. We quantified the Spearman's rank correlation between each variable's raw value and the variable's SHAP value. We additionally performed the above analysis for all species as reported in the supporting materials.

**(ii) Deconstructing distributions to build conservation expectations.** By decomposing environmental suitability into component variable contributions, calculating SHAP values enabled us to define our properties of species' distributions: '*expected distributions*' and '*shadow distributions*' (Fig. 1). We calculate these distributions as a set using a binary form and as properties of this set using quantitative representations detailed below. We provide a conceptual overview of these in Fig. 1 and code to reproduce these properties in Supplementary Note 2.

**Expected distributions.** We define a binary expected distribution as the set of sites (here sub-catchments) inside the abiotic niche of a species (i.e., separate from any consideration of threats; Fig. 1). Here, we assumed the factors discharge, slope, temperature, flow velocity, and distance to lakes contributed to species' abiotic environmental niche and represent "natural" ecological constraints on species distributions. We define the "binary expected distribution" as:

$$binary\ expected\ distribution = E^b := \left\{ s_i\ \forall\ i \in S: \sum_{\{n \in N\}} SHAP_{in} > 0 \right\}, \quad (1)$$

where $s_i$ is the $i$ th site in the set of all sites $S$, and $N$ is the set of natural variables. This gave a reference set of sub-catchments describing whether sub-catchments were inside or outside of the expected distribution (i.e., the naturally realised abiotic niche) for each species.

We define a property of each site inside the set defined by the binary expected distribution and call this the species' "quantitative expected distribution", defined as:

$$quantitative\ expected\ distribution = E_i^q = \sum_{\{n \in N\}} SHAP_{in} + \sum_{\{a \in A\}} Q_{0.95}(SHAP_a) + \hat{y}, \quad (2)$$

where $\hat{y}$ is the model mean predicted habitat suitability across all sites, and $A$ is the set of anthropogenic threat variables including urbanisation, river morphological modification index, (reduction of) floodplain

area, (reduction of) wetland area, and (loss of) connectivity. This property represents the improvement towards the optimal condition of a location for each individual species when alleviating a threat. We simulated the alleviation of threats, as a best-case scenario by calculating the 95th quantile of threats positive SHAP values. We used the 95th quantile of SHAP values to avoid spuriously large positive SHAP values affecting our measure of the maximum.

**Shadow distributions.** We also define binary and quantitative properties of a species' *shadow distribution*, which give different insights into negative anthropogenic influences inside species expected distributions (Fig. 1). We first define the *binary shadow distribution* as:

$$binary\ shadow\ distribution := \left\{ s_i \ \forall \ i \ \in \ S : E_i^b, \sum_{\{a \in A\}} SHAP_{ia} < 0 \right\} \quad (3)$$

We calculate this *binary shadow distribution* for different sets of $A$: for each threat, combinations of all threats, and subsets of different threats. We combined indicators of habitat-loss related threats of (low) floodplain cover, (low) wetlands cover, (high) river morphological modification index, and (high) urbanisation into an indication of 'habitat quality' and perceive reduced habitat quality as a threat to species' populations. We present this measure in the main manuscript for *Alburnoides bipunctatus*.

We also defined a *quantitative shadow distribution* as:

$$quantitative\ shadow\ distribution = \text{if} \left\{ s_i \ \forall \ i \ \in \ S : E_i^b, \sum_{a \in A} SHAP_{ia} < 0 \right\}, \text{then } \frac{y_i}{E_i^q}, \quad (4)$$

where $y_i$ is the site's environmental suitability score. The *quantitative shadow distribution* estimates the fraction of environmental suitability loss due to human impacts inside the abiotic niche of species. For consistency, we calculated $y_i$ inside the SHAP framework by adding the mean environmental suitability score to the local sum of SHAP values, giving the site environmental suitability score (but is equivalent to the random forest model prediction).

We calculated the above properties for all species and estimated across all species per sub-catchment the average environmental suitability for observed distributions and expected distributions. In addition, we calculated the mean, minimum and standard deviation in quantitative shadow distribution across species. We also averaged the presence of negative influences for habitat quality SHAP values (defined above) and connectivity loss SHAP values as anthropogenic threats. For example, where three of four habitat quality threats negatively impact a species in a sub-catchment, the catchment received a score of three.

Our estimation of shadow distributions by adjusting SHAP values (e.g., $Q_{0.95}(SHAP)$) is a hypothetical scenario and comes with assumptions and uncertainty. To understand the impact of these choices on our results, we generated two other hypothetical threat alleviation scenarios. We calculated a very conservative scenario by converting negative SHAP values to 0, in this scenario threats no longer have a negative contribution to environmental suitability (but the underlying factor also does not contribute positively to environmental suitability). Second, we converted negative SHAP values to the mean positive SHAP values for each threat factor, which indicates a positive recovery of threats to the average condition in unthreatened regions for each threat factor. In addition to our SHAP adjustment, we tested an approach to estimate shadow distributions where we adjust the feature values in the environmental data directly and compare the observed and expected distribution of suitability scores (see Supplementary Fig. 6). This approach simulates improvements in environmental states and makes new predictions given these improvements. In this approach, we replaced environmental values of threat features

to be the 99th quantile if a high value represents an improved state (such as higher connectivity) or 1st quantile in the inverse case, such as lower morphological modification. We found the output from this non-SHAP method to be very highly correlated to the SHAP method presented in the main manuscript for estimating shadow distributions (median correlation across species = 0.88, IQR = 0.85–0.89; Supplementary Fig. 6). For consistency, we present here only the first described SHAP based shadow distributions described in Eq. 4, but note that shadow distributions, like geographic ranges, are latent properties, so perfect calculation is impossible and estimation methods are required.

**(iii) Local relative contributions between contrasting locations.** We used SHAP values to identify the relative local contributions of each variable to inform which environmental factors, at a management scale, support or decrease environmental suitability for a species. We apply these insights across our nine focal species in two contrasting river systems: the sub-catchments comprising the main stems of the upper Emme River (32.8 km$^2$) and Sense River (35.1 km$^2$). We chose these catchments intentionally to potentially form contrasting case studies, given our on-site knowledge. These rivers are qualitatively similar in terms of abiotic environments (e.g., discharge, temperature, and flow velocity). However, the upper Emme is heavily modified in some sections for flood prevention and has downstream run-of-river hydropower production since the late 1800s leading to historically low connectivity. The Sense has a higher degree of connectivity with a more natural and largely unmodified flow regime (Supplementary Fig. 7). We calculated the mean SHAP values per river catchment per species and compared these values between rivers and species.

The data to reproduce this work are available at https://doi.org/10.6084/m9.figshare.24787227[86], and the code associated with reproducing the analysis and figures in this manuscript are available at https://doi.org/10.5281/zenodo.13626649[87].

### Reporting summary

Further information on research design is available in the Nature Portfolio Reporting Summary linked to this article.

## Data availability

The data to reproduce this work are available at https://doi.org/10.6084/m9.figshare.24787227[86], which contains Supplementary Data Files 1-3 as well as scripts to reproduce our underlying SDMs and SHAP analysis. Supplementary Data File 1 contains outputs of species distribution models and SHAP analysis for *A. bipunctatus* and underlying data layers to reproduce Figs. 2, 4. Supplementary Data File 2 contains outputs of SHAP analysis for all species in our analysis underlying data layers to reproduce Figs. 3, 7. Supplementary Data File 3 contains calculated shadow distributions and expected distributions, with underlying SHAP analysis for all species, and all underlying data layers to reproduce Figs. 5, 6.

## Code availability

The code associated with reproducing the analysis and figures in this manuscript are available at https://doi.org/10.5281/zenodo.13626649[87] with the full code pipeline to fit SDMs also available at https://doi.org/10.6084/m9.figshare.24787227[86].

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

## Acknowledgements

This work is part of the project LANAT-3 'Den Biodiversitätsverlust der Gewässer stoppen — trotz Klimawandel' funded by the Wyss Academy for Nature through the implementation programme with the canton of Bern (Office for Agriculture and Nature) and by the Federal Office for the Environment (FOEN). This project funding supports C.W., B.W., D.J., and B.C. Thanks to the LANAT-3 team and the advisory boards for project feedback. Thanks to Dr. Pascal Vonlanthen at Aquabios and Dr. Sebastien Lauper at Canton Fribourg for providing MSK data. Thanks to Michael Häberli for providing Canton Bern monitoring data. Thanks to Dr. Rosi Siber for providing advice on environmental data. Thanks to Sophie Moreau, Anita Schmid, Hiranya Sudasinghe, Marion Talbi and Ian Woodman for fieldwork support and especially Marcel Häsler for project and fieldwork support. Thanks to the EAWAG Department of Fish Ecology & Evolution for thoughtful discussions and feedback. Special thanks to Dr. Carlos Melian for fruitful discussions on the conceptual and mathematical representation of shadow distributions. Thanks to Lukas Rüber and Soraya Villalba for support with specimen and database curation. Calculations were performed on UBELIX

(http://www.id.unibe.ch/hpc), the HPC cluster at the University of Bern.

## Author contributions

Author contributions following the Contributor Roles Taxonomy (CRediT): C.W. – conceptualisation, methodology, software, validation, formal analysis, investigation, data curation, writing – original draft, visualisation. B.W. and B.C. – validation, investigation, data curation, writing – review and editing. D.J. – investigation, data curation, writing – review and editing, project administration. L.J.Q. – data curation, writing – review and editing. J.B. – investigation, resources, data curation, writing – review and editing, O.S. – conceptualisation, validation, investigation, resources, writing – review and editing, supervision, project administration, funding acquisition.

## Competing interests

The authors declare no competing interests.
