## [Peer Review File · Nature Communications]

Deconstructing the geography of human impacts on species' natural distributionREVIEWER COMMENTS

Reviewer #1 (Remarks to the Author):

I have reviewed the manuscript entitled "Shadow distributions: Deconstructing the geography of human impacts on species' natural distribution." The study aims to quantify the relative contribution of multiple environmental factors that affect species' habitat suitability at each location across species' geographic distributions, using a species distribution modeling (SDM) approach. To do so, the study attempted to differentiate the effects of natural factors (undisturbed state) and those of anthropogenic factors. The modeling exercise was demonstrated for nine freshwater fish species with 11 covariates in Switzerland. The challenge is an important topic in ecology, the manuscript is written clearly, and the method is relatively new in the ecology domain. However, I do not think that this study brought substantial progress to be reported in Nature Communications. Also, there are some important concerns.

This study applies extensively the popular explainable AI (XAI) method, SHAP (SHapley Additive exPlanation). It is a nice use case to demonstrate how SHAP can be used to elaborate what a machine learning (ML) algorithm learned from the data. I can admire that the authors did a nice modeling work to elaborate on what XAI can do for SDM. I can imagine this manuscript can be a good candidate for a top journal in the discipline (e.g. ecography).

Yet, I found two problems about SHAP: (1) the study does not report model performances. SHAP is a post hoc explanation method, and therefore, if the fitted model performs poorly, it is meaningless to explain the low-performance model. (2) The method section continuously mentions SHAP as shapley, but SHAP is an extension of shapley. Shapley values are about coalition game theory, but SHAP uses the idea of shapley game theory to estimate shap values for predictors used in the model. The authors may want to read e.g. Molnar's book "Interpreting Machine Learning Models With SHAP" to better understand the methods. It also helps them to more clearly describe which SHAP estimate they used.

The introduction section is misleading. The section emphasizes the level of "population" but their study is not about population. An SDM approach uses only presence-absence (binary) data, which does not necessarily represent the population dynamic of a species (like metapopulation processes incl. local competition and regional dispersal). I believe that the study wants to argue mainly about "habitat suitability (or the probability of presence)".

Therefore, I see the definition of "shadow distribution" as problematic. It is defined as "the area where natural niche factors positively contribute to species population performance, but threats, contributing negatively, reduce population performance — quantifying the extent that an observed species distribution is in the shadow of human influence"  please note that predicting presence probability is not about population performance (i.e. fitness) but habitat suitability. Also, there is no word like "natural niche factors" (you can try to google it). It is nice to carefully check terms used consistently in the SDM domain.

The authors assumed stream discharge, temperature, and flow velocity are "natural factors" (undisturbed) with a justification as "they could not be modified by human actions." But, there are hundreds of stream ecology studies about the human impacts on these factors (related to

environmental flow, flow regimes, and thermal regimes modified by water use and artificial dam operation -- e.g. a lot done by N. LeRoy Poff, Angela Arthington, Julian Olden). A more careful justification and reasoning needed.

Reviewer #2 (Remarks to the Author):

Thank you to the authors and editors for the opportunity to read this interesting manuscript. My review is below.

—Summary of Article—

The authors investigate the use of Shapley values to characterize the local drivers of habitat suitability and anthropogenic threat experienced by a species. They coin the term “shadow distribution” for the portions of a species’ (real or potential) range which could be suitable for the species were it not for the effect of anthropogenic threats. They explore this approach using a selection of 9 freshwater fish in Switzerland and a random forest SDM over subcatchments based on a suite of 18 potential covariates. This approach is used to dissect the differing species responses to environmental covariates, to inspect the spatial distribution of this variance, and the potential of a spatially explicit understanding of the drivers of species suitability to direct conservation decision making.

—Summary of Review—

As the authors claim, the application of XAI to SDMs is a natural extension of prior work on variable importance in SDMs and a potentially useful contribution to the effort to separate a species’ fundamental and realized niche. However, I believe more significant attention should be given in the main text to the definition and interpretation of Shapley values in order to draw a clearer line between their statistical and causal interpretations, particularly as they are related to conservation decision making and to the concept of a “shadow distribution”. That said, I am not aware of any existing literature that has presented these ideas and my technical concerns about the methods here are minor, so I recommend revisions to the paper need only be relatively minor.

—Significant Feedback—

- The article should be related better to existing variable importance work in SDMs. Specifically, how does a site-specific breakdown of variable importance from Shapley values compare to range-wide variable importance measures in random forests or other single-species SDMs? Ideally this would be explored in some of the case studies presented (i.e. comparing Shapley values w/ a species-wide measure), but additional context is necessary regardless in order to understand its relevance to conservation.
- Some additional background on the definition and interpretation of Shapley values should be added to the main text. I appreciated and support the authors’ more technical presentation of Shapley values in Appendix 3, and found the included RMD vignette interesting, but found what was kept in the main text to do an insufficient job, especially for an audience potentially unused to variable importance in general and likely unfamiliar with XAI or locally-interpretable attribution measures. What is presented in the main text is of potentially limited utility to an

unfamiliar reader (ex. what do the 4 game-theoretic optimality conditions of Shapley values have to do with interpreting environmental suitability/threats?). I suggest adding a conceptual figure (perhaps based on some of the material in Appendix 4?) and expanding the section in the main text so that a reader who reads only the main text will have a serviceable understanding of XAI and Shapley values.

- Several of the manipulations of Shapley values are new to me and feel worryingly unusual. Specifically, the correlations of Shapley values w/ covariates, stacking of Shapley values to define a “shadow” distribution, and the hypothetical exploration of the effect of “mitigating” threats by modifying features according to their Shapley values. I suggest these concepts could be more well supported in the main text or appendices, as they get away from traditional uses of Shapley values in ways that are potentially unique and powerful.

- An additional note should be added to the cautions and limitations section on the causal misinterpretation of Shapley values. I believe this is especially important given the experiments varying features to more “ideal” values wrt their Shapley values and the relation to conservation planning. Of course, Shapley values can provide useful insights to planners and managers, but readers should probably not be given the impression that Shapley values give us levers to control directly and quantitatively in habitat restoration efforts.

—Minor Feedback—

- Much of the details on the reduction from 18 covariates to 11 could be moved to an appendix, especially to keep focus on the novel variable importance methods

- Similarly, the coupling/decoupling and convergence/divergence analyses were interesting, but could also move to an appendix

—Line Items—

L38-44: these lines don't scan well to me. Much of the intro could use some refocusing (e.g. also L72-77)

L172-176: confusingly worded

L185: be more specific about what “successfully removed” means (or cut some of this material, as suggested above)

L204: exAI → XAI

L272-276: doesn't scan

L304: what are “connectivity threats”?

L408: missing “of” in “14% sub-catchments”

L602: typo: “Previous largely”

L606: typo: threat → threats

Reviewer #3 (Remarks to the Author):

The authors present a study on (anthropogenic) threat mapping of freshwater species and disentangling from natural environmental factors, resorting to correlative distribution, resp. habitat suitability modelling, followed by explainable AI (XAI) methods via Shapley additive values (SHAP) to identify contributing factors. The study overall appears diligently laid out, well thought through, and well executed. The methodology itself is not new, nor is it entirely solid (see comments below, especially regarding potential variance inflation). However, this is one of the first papers applying it in a fully proper way, with attention to resolving unwanted artefacts like collinearity. The key contributions of the work are twofold, including (i.) an introduction of a new concept ("shadow distribution"), and (ii.) end-to-end application and case study on well-monitored ecosystem and species.

Thus, this work certainly is of great value and worth publishing, upon having answered a few concerns I have. Please see major and minor comments below.

MAJOR COMMENTS

- The Introduction section is cumbersome to read and not entirely insightful. It feels as if the authors tried to fully separate methodology from concepts, but have gone a bit too far—for example, the present work basically builds on species distribution modelling (SDM), but that large branch of research is not even mentioned until later. The inclusion of the Kunming framework is not done sufficiently—any study on environmental drivers could help the framework, but this work here can do so in a different way by looking at local effects. There are strongly supporting passages in the framework regarding this topic, which are currently unaddressed.

- I am missing a critical reflection on the limitations of the methodological approach itself. While the section on "Cautions and limitations" (p.20f, l.609ff) talks about pitfalls in applying the methodology, there is nothing about its inherent shortcomings. These include (among others) the following:

a. Risk of variance inflation (p.7, l.240ff): It conceptually makes sense to relate covariates values to their local importance scores. However, you are doing this on (i.) Shapley values, which are a statistically uncertain model (and even worse so as you have to approximate), (ii.) RF predictions (yet another statistical model), and (iii.) residuals instead of measured values for some covariates emerging from a GAM (yet another one). You do so using a Spearman correlation (another statistical model). This sounds like a recipe for trouble due to variance inflation. My take is that while you certainly can do that, results should be interpreted with utmost care, as the likelihood of encountering a measurement that is too uncertain to be trusted, or worse, spurious correlation artefacts, is very high.

b. Shapley values require input covariates to be uncorrelated. The authors address this in a good way by de-correlating them via a GAM and regression splines and using only residuals for some variables (although they do so for facilitated interpretability reasons and never mention this requirement). It is obvious that this introduces yet another level of uncertainty (model choice and parameterisation), and that this is not the only possible solution to this end.

c. Shapley values themselves are not the only means for XAI. The authors briefly mention one competing method (LIME) and cite other papers on why their method is to be preferred. This by itself is not wrong; Shapley values have shown to be very robust and less susceptible to

unwelcome error sources indeed; it is also okay to not elaborate too much on the technical terms here, as this is not the key focus of the paper. However, this likewise has to be mentioned as a potential error source.

- The Conclusion section is detached from the antecedent text and underwhelming. Please pick up again the gist/major messages from the paper.

MINOR COMMENTS

- p.1, l.27: "widely used threat and biodiversity mapping exercises" → too vague
- p.2, l.35ff: Would it be possible to add an example on such "specific threats" (i.e., elaborate on why a "simple" expansion of protected areas does not always suffice)?
- p.2, l.44ff: A mention of the Kunming GBF is of course almost a given here, but I would recommend providing a bit more detail (e.g., highlight some of the goals of the framework that specify and rely on local conservation actions). Otherwise, this could just as well be mentioned for global studies (it currently does not add much to the "local" argument).
- p.2, l.47: Consider replacing "downscaling"; this has an unjust negative connotation here.
- p.2, l.60f: Rather strange sentence ("when mapping and ranking threats, researchers assume that threats are important (...)"). Consider reformulation.
- p.4, l.126f: what kind of "modern threats" are potentially absent from data of 2010 and earlier? In other words, can you explain in more detail why you limited measures to 2011 and newer?
- p.4, l.131f: Did you aggregate observations in time? Would it make sense to conduct analyses with time as a factor? I could imagine threats having changed over time, for example due to the Covid-19 pandemic.
- p.5, l.149: Did you mean "local values *within* 100m from river"?
- p.5, l.157: What do you mean by "downsampled" random forests?
- p.5, l.159f: "(...) and perform as well as model ensembles" → Sort of illogical clause. The performance of RF heavily depends on the experimental setup and may not always be better than an ensemble of models; also, strictly speaking, RF already is an ensemble (of bootstrapped decision trees). Consider dropping clause.
- p.5, l.161f: Class imbalance does not lead to overfitting, but to biased predictions. The concept of overfitting pertains to performance issues of a converged model when deployed on out-of-training set data. Also, downsampling is a heavily compromised approach to counteract imbalances as it drops potentially vital training points; did you consider other remedies?
- p.5, l.163: what was the "balanced subsample" size (how many presence/absence points per decision tree)?
- p.5, l.163ff: Strictly speaking, RF also do not provide occurrence probabilities in unmodified settings; all they report is the fraction of agreement across the trees. No action needed; I approve of the term "environmental suitability".
- p.5, l.167f: Did you conduct a grid search, resp. hyperparameter tuning stage, or are these parameters just set by hand? Also, please refrain from referring to "default" parameter values, as a default is generally implementation-specific.
- p.5, l.169: You are addressing the elephant in the room here with multicollinearity. This is not (just) because of the SDM (RF can work very well even if covariates are collinear) and importance interpretation, but because of the Shapley values framework itself you use for XAI. A vital requirement for these is covariate independence, a property often ignored by existing studies. I strongly recommend mentioning this requirement (with a reference) at an appropriate

stage in the manuscript.

- p.5, l.176ff: I really like the idea of using GAM residuals to test for confounders. Since RF can model nonlinear relationships, it makes sense to test for nonlinear collinearity effects as well here. I assume this is why you used regression splines. I am wondering though how much effect a different choice of data transformation and/or link function in the GAM could have on the outcome of the collinearity analysis. Did you try other nonlinear variants to see what the outcome is?

Aside, if you mention the software used (mgcv), please do so as well for random forest.

- p.5, l.179ff: Again, using residuals as input is an intriguing idea. It does sound logical, but its accuracy all (again) hinges on the quality of the GAM used. This could benefit from additional analyses.

- p.6, l.200: What do you mean by "final random forests"? Did you use the five-fold CV for hyperparameter tuning (e.g., no. trees, which covariates to use) and had a separate, held-out test set, or for testing of the final model? → Unclear.

- p.6, l.202ff: I would mention the term "local feature importance" here explicitly, as it is common and has been around for quite some time in the machine learning community.

- p.6, l.204: "XAI" is a more common acronym for the concept than "exAI" I believe.

- p.6, l.214f: This statement simply is not true. There are many more references beside the three mentioned ones, such as:

Cha, Y., Shin, J., Go, B., Lee, D.S., Kim, Y., Kim, T. and Park, Y.S., 2021. An interpretable machine learning method for supporting ecosystem management: Application to species distribution models of freshwater macroinvertebrates. *Journal of Environmental Management*, 291, p.112719.

Bourhis, Y., Bell, J.R., Shortall, C.R., Kunin, W.E. and Milne, A.E., 2023. Explainable neural networks for trait-based multispecies distribution modelling—A case study with butterflies and moths. *Methods in Ecology and Evolution*.

He, B., Zhao, Y. and Mao, W., 2022. Explainable artificial intelligence reveals environmental constraints in seagrass distribution. *Ecological Indicators*, 144, p.109523.

Miyaji, R.O., Almeida, F.V. and Corrêa, P.L.P., 2023, September. Evaluating the Explainability of Machine Learning Classifiers: A case study of Species Distribution Modeling in the Amazon. In *Anais do XI Symposium on Knowledge Discovery, Mining and Learning* (pp. 49-56). SBC.

Ryo, M. and Angelov, B., High accuracy is not enough: Interpretability in species distribution model increases with explainable artificial intelligence/interpretable machine learning.

- p.7, l.258ff: I understand the motivation behind this, but what do you do if a variable is an abiotic, contributing factor and *can* be directly influenced by human actions?

- p.7, l.259ff: This in turn I am not sure about. Are Shapley values really comparable across covariates? My intuition tells me that this is true if the covariates are normalised (e.g., by Z-scoring). However, you are also feeding residuals into the model. This merits further investigation. The same goes for the threat alleviation setup following right after.

- p.8, l.296ff: I've been expecting the shadow distribution to be defined as the difference between expected and observed. This intuitively makes sense—but only if the expected distribution can factor out *all* disturbing factors. This relates to the discrepancy between what is observed regarding species (presence/absence) and what is measured with respect to environmental factors; in a nutshell, it boils down to the problem that observations exhibit the realised, and measurements the fundamental niche. This is an age-old conundrum in SDM works. Now, my key concern lies in the fact that the realised niche is not just influenced by abiotic drivers and human disturbances (as modelled by you), but other, non-considered factors as well, including any type of biotic interaction. Hence, even though you make a good effort in

defining disturbance factors, your "shadow distribution" is nonetheless confounded by ignored factors like biotic ones, and should be interpreted with care.

- p.10, Figure 1: Multiple issues:

- Panel (b): Could you add whiskers to the bar chart indicating e.g. the standard deviation of each average importance?

- The individual graphs of the subpanels are rather hard to read.

- Resolution is low. Do you have a better quality version of it, ideally with vectorised textual descriptions?

- p.10, Figure 1: Moreover, I believe parts of this figure to exemplify the limitations I see with the approach. Let us consider discharge—unsurprisingly, values are highest directly measured over major rivers (panel (e)). Just as logically, occurrence of the species is highest in rivers (panel (a)). Unsurprisingly, river discharge then emerges as the most important contributing factor (panels (b), (c)). However, the vast majority of this explanatory effect just states the obvious (i.e., fish don't naturally occur outside water bodies). Worse, it may masquerade bias effects you may not have considered—for example, RF may predict highest suitability in the biggest rivers (Aare, etc.), simply because the species was observed more there. Accordingly, discharge again emerges as the biggest contributing factor, because these rivers are large. What if a certain species actually finds better conditions in smaller tributary rivers, but this just does not emerge because of sampling biases? → In sum, XAI methods suffer from precisely the same biases and issues as prediction models (SDMs) in the first place.

- p.11, l.360ff: Indeed; this is the root of it all. Do you by any chance have examples from the literature on how your focal species react to threats? If another study found that e.g. *Gobio gobio* was more sensitive to urbanisation than *Cottus gobio*, for example, and you could replicate this hypothesis with your approach, it would greatly add value to your work!

- p.11, l.368f: This is just as interesting and could lead to lots of work regarding e.g. invasive species.

- p.12, Figure 2: Great figure! I initially was a bit skeptical about comparing species even though they were fitted using independent models, but since the sampling design is the same and biotic interactions are explicitly left out, this should not cause problems.

I believe the side captions on the right for (iii.) and (iv.) are reversed, compared to the figure caption.

- p.13, Figure 3: This figure is hard to read in monochrome/red-green blindness. Can you use a better colour ramp, or even better, also add different shapes to the points? Also, maybe highlight *O. mykiss* explicitly since it's the non-native one. Speaking of which, I would love to see some discussion on this species' response to its outlier positions in particular, such as flow velocity and morphological modification index.

- p.15, Figure 4: Again, very interesting figure! One minor question of interest: you explained how you calculated habitat quality above in the text (as a combination of multiple factors). Here, I see poor habitat in some areas in the central-north part of the country, but also in many other places together with poor connectivity (panel (c)). Can you provide further information on what exactly it is that makes these regions unsuitable habitats?

- p.18, l.505: I would be careful with weighting terms like "successful" in this context.

- p.18, l.513ff: This second contribution isn't all that new; I give credit that you were the first to properly address it in an SDM context (with sensible heeding of statistical requirements, such as variable independence) and with separation of natural/nonnatural threats.

- p.19, l.545ff: Sure, but the term "shadow distribution" first needs to be accepted by the community. Consider down-toning statement and targeting the idea, rather than the term you coined yourselves.

- p.19, l.551f: You are crossing scales and mapping units by a lot here (from regional to global). While the application of your framework in global studies may work, it is not clear how trustworthy results become then, and how well drivers can be separated with respect to latitudinal gradients.
- p.20, l.596ff: I am not sure I fully see through your line of argumentation here. Those two postulated sources (species can respond differently to environmental gradients and vice versa) sound rather obvious and not really related to the study in full. Also, there's a linguistic problem here (facsimile: "two major sources of variation contributing to responses: i) species having different responses")—the source of variation in response is differences in response...
- p.20, l.599f: Above you claimed the opposite (w.r.t. latitudinal gradient).
- p.20, l.602: Not really. Location-specific XAI methods have been around for a long time; I don't see this as your primary contribution. Rather, your study is the first to study it in-depth in an SDM context and the first to provide further-reaching analyses w.r.t. disturbances (as opposed to just environmental factors).
- p.20, l.609ff: While it is common practice to place a limitations section at the end, I strongly advise adding some limitations above, perhaps already when introducing the methods. As said in one of the major comments above, there are quite some more pitfalls regarding your methodology than you write about here.
- p.20, l.612ff: You do have a point here, but there is a circularity to the argument: precisely because many large and/or multi-species models are tuned for accuracy, XAI methods like Shapley values are often a first step towards actually understanding the response curves.
- p.20f, l.615ff: I don't like this as a general statement, as not all applications require explainability over high accuracy.

Reviewer #3 (Remarks on code availability):

Disclaimer: I have not run the code (and thus ticked "No" on the code review question), but have read through it.

Code is at academic/scientific level (not enterprise-grade) but seems legible and intuitive. Supplemented walkthrough document (Appendix 4) is excellent and explains code step-by-step. No major blunders detected.

RESPONSE TO REVIEWERS' COMMENTS

Reviewer #1 (Remarks to the Author):

I have reviewed the manuscript entitled "Shadow distributions: Deconstructing the geography of human impacts on species' natural distribution." The study aims to quantify the relative contribution of multiple environmental factors that affect species' habitat suitability at each location across species' geographic distributions, using a species distribution modeling (SDM) approach. To do so, the study attempted to differentiate the effects of natural factors (undisturbed state) and those of anthropogenic factors. The modeling exercise was demonstrated for nine freshwater fish species with 11 covariates in Switzerland. The challenge is an important topic in ecology, the manuscript is written clearly, and the method is relatively new in the ecology domain. However, I do not think that this study brought substantial progress to be reported in Nature Communications. Also, there are some important concerns.

This study applies extensively the popular explainable AI (XAI) method, SHAP (SHapley Additive exPlanation). It is a nice use case to demonstrate how SHAP can be used to elaborate what a machine learning (ML) algorithm learned from the data. I can admire that the authors did a nice modeling work to elaborate on what XAI can do for SDM. I can imagine this manuscript can be a good candidate for a top journal in the discipline (e.g. ecography).

We thank the reviewer for their constructive comments.

Yet, I found two problems about SHAP: (1) the study does not report model performances. SHAP is a post hoc explanation method, and therefore, if the fitted model performs poorly, it is meaningless to explain the low-performance model.

We state on lines 222-230 that we report model performances using 13 metrics of model performance. In the main text we refer to the ODMAP protocol for more details Figure S4, Table S2 and S3 in Appendix 2, which provides quite an exhaustive overview of model performance. We also already provided comparisons of multiple sensitivity tests across different data subsets to check the validity of our models to data inputs (Figure S1 and S2). We believe these validations are beyond what most applications of species distribution models check in their model performance. For threshold dependent metrics we also provide two types of thresholds based on both the True-Skill Statistic and Matthews Correlation Coefficient. Apart from the below two species, performance based on AUC averaged 0.91 ± 0.02 . Two species have weaker models based on AUC metrics for well-known reasons that justify their inclusion: *Cottus gobio* (AUC = 0.70) have very high prevalence in Switzerland, which is well known to reduce model performance (Lobo et al. 2008 but see Lawson et al. 2014), and *O. mykiss* (AUC = 0.66) is a non-native species with a distribution almost exclusively determined by stocking which decouples the realized from fundamental niche and enables species overcome natural dispersal barriers. We think it interesting to include *O. mykiss* as a contrastive example from a non-native species, as reviewer 3 also highlights.

We now also highlight that well-performing models are important for model interpretability in the discussion on lines 692-694 when incorporating the other limitations highlighted from reviewer 3's comments.

Lawson, C.R., Hodgson, J.A., Wilson, R.J. & Richards, S.A. (2014). Prevalence, thresholds and the performance of presence-absence models. *Methods in Ecology and Evolution*, 5, 54–64.

Lobo, J.M., Jiménez-Valverde, A. & Real, R. (2008) AUC: a misleading measure of the performance of predictive distribution models. *Global Ecology and Biogeography*, 17, 145–151.

(2) The method section continuously mentions SHAP as shapley, but SHAP is an extension of shapley. Shapley values are about coalition game theory, but SHAP uses the idea of shapley game theory to

estimate shap values for predictors used in the model. The authors may want to read e.g. Molnar's book "Interpreting Machine Learning Models With SHAP" to better understand the methods. It also helps them to more clearly describe which SHAP estimate they used.

We have replaced all instances of 'Shapley' value with SHAP value as suggested and based on the terminology described in Molnar's "Interpreting Machine Learning Models With SHAP" page 19. This suggests using "Shapley values" describes the original method from game theory, SHAP is the application of Shapley values for interpreting machine learning predictions and "SHAP values" refer to the resulting values from using SHAP on model features.

We now state this on lines 234-236: *"To do so, we approximate Shapley values defined in game coalition theory⁴⁵ using SHapley Additive exPlanations which is known as "SHAP" when applied to explain predictions from machine learning models⁴⁵⁻⁴⁸".*

The introduction section is misleading. The section emphasizes the level of "population" but their study is not about population. An SDM approach uses only presence-absence (binary) data, which does not necessarily represent the population dynamic of a species (like metapopulation processes incl. local competition and regional dispersal). I believe that the study wants to argue mainly about "habitat suitability (or the probability of presence)".

Therefore, I see the definition of "shadow distribution" as problematic. It is defined as "the area where natural niche factors positively contribute to species population performance, but threats, contributing negatively, reduce population performance — quantifying the extent that an observed species distribution is in the shadow of human influence"  please note that predicting presence probability is not about population performance (i.e. fitness) but habitat suitability.

We clarify that niche based theories that underly species distribution modelling are based on ideas of how individual and population level "fitness" or "performance" vary across environmental gradients (lines 54-65). We argue below why it is valid to reference the population level in our introduction. We believe the introduction is not misleading but instead echoes classic work on ecological niches (Hutchinson 1957; Hutchinson 1978; Colwell 2009; Holt 2009; Soberon 2010). In this foundational work on ecological niches, population performance along environmental gradients is a fundamental concept helping define species niches. It must be noted that SDMs are a subfield of study that *applies* these theories to geographic range structure and ecological niche theory. We prefer to make the definition of a theoretical biological/ecological property (expected and shadow distribution) independent of the tool or method (SDMs + SHAP) used to estimate the property (however, as requested by reviewer 3 we also introduce SDMs now in the introduction). Habitat suitability is a restrictive term used almost exclusively to define the index of predictions from a species distribution model. However, we believe shadow distributions should not only be defined by habitat suitability scores, but could also be defined by multiple ecological response variables at population-to-species level of biological organisation. For example, it could also be measured as the regions within a species range where the intrinsic rate of population growth is < 0 due to anthropogenic threats, or where abundance is reduced.

Given the comments that populations are not relevant to our paper, we emphasise below how the theoretical foundation of species distribution modelling arises from an attempt to quantify a species niche through population variation in abundance or occurrence (beyond how this is implemented in SDMs). The origins of the concept of a multidimensional ecological niche comes from Hutchinson proposing a mapping of population dynamics onto environmental space (Hutchinson 1957) which is later built on in "An Introduction to Population Ecology" (Hutchinson 1978). Along this line, modern conceptual papers highlight the fundamental role that environments have in constraining population level processes of births, deaths, immigration and emigration (e.g., demographic rates) which then

defines the ecological niche of a species (Holt 2009; Soberon 2010). As such, we can view the niche of the species as the set of environments where the intrinsic rate of population growth (r) is > 0 , and the geographic manifestation of this niche as the environments that enable an $r > 0$ (Colwell 2009). Given that demographic rates and r are extremely challenging to quantify across broad areas of species distributions, the use of proxy variables such as population abundance and most frequently the presence of species, has become common to represent the distribution of species in geographic space (with SDM users perhaps sometimes forgetting the underlying population theory) and to quantify the resulting ecological niches of species through models such as SDMs. So, while pragmatism means that the real data being modelled is the presence or absence of individuals, the underlying theoretical concept we are attempting to quantify is the mean fitness a populations across environmental space. There are many studies picking apart whether such widely held assumptions on the mapping between SDMs and biological properties of populations are robust (Vanderwal 2009; Dallas et al. 2018; Lee Yaw 2019), but it is beyond the scope of our manuscript to deal with these issues. Given the above, we believe it not only acceptable, but preferable, to relate our concepts to population processes rather than exclusively use the term habitat suitability in the introduction, which conflates the tool used in a subfield of ecological niche and geographic distribution theory (SDMs) with the theoretical niche concepts that are founded in population biology. We now cite the below literature (except Hutchinson 1978) in our introduction.

Colwell, R.K. & Rangel, T.F. (2009). Hutchinson's duality: The once and future niche. *Proceedings of the National Academy of Sciences*, 106, 19651–19658.

Hutchinson GE (1957) Concluding remarks. *Cold Spring Harbor Symp* 22:415– 427.

Hutchinson, G.E. (1978). *An introduction to population ecology*. Yale University Press.

Holt, R.D. (2009). Bringing the Hutchinsonian niche into the 21st century: Ecological and evolutionary perspectives. *Proceedings of the National Academy of Sciences*, 106, 19659–19665.

Soberón, J.M. (2010). Niche and area of distribution modeling: a population ecology perspective. *Ecography*, 33, 159–167.

Also, there is no word like "natural niche factors" (you can try to google it). It is nice to carefully check terms used consistently in the SDM domain.

We have rephrased all instances of “natural niche factors” to “natural abiotic factors defining the realized niche of the species” and “natural niche” to “abiotic realized niche”. Although not asked for, we now emphasise the realized niche of the species because, as with most SDM applications, we cannot fully control for biotic factors that jointly determine species distributions.

The authors assumed stream discharge, temperature, and flow velocity are "natural factors" (undisturbed) with a justification as "they could not be modified by human actions." But, there are hundreds of stream ecology studies about the human impacts on these factors (related to environmental flow, flow regimes, and thermal regimes modified by water use and artificial dam operation -- e.g. a lot done by N. LeRoy Poff, Angela Arthington, Julian Olden). A more careful justificatoin and reasoning needed.

We have now clarified our meaning of natural factors in the manuscript and refer to the ODMAP protocol (Table 3 in Appendix 1) for more explicit definitions of all variables. We are aware of the many studies showing how discharge, flow etc can be modified by humans. However, the metrics of discharge, temperature and flow velocity that are available in a spatially continuous format required for our study do not capture these anthropic modifications. Instead, our data are coarser scale variables that represent a natural river continuum from small, fast flowing, steep cold headwater streams to larger, slower flowing warmer main-stems of rivers. We now have clarified the above in the methods stating on lines 297-303:

“We assumed that the natural factors contributing to the abiotic environmental niche were discharge, slope, temperature, flow velocity, and distance to lakes. Our derivation of these variables is not fine-scale enough to incorporate well-known human influences on river temperature, flow or discharge regimes but instead represent gradients along the natural river continuum (cold, fast flowing, small headwater streams to warm, slow flowing, large main-stems), and we assume these factors represent natural ecological constraints on species distributions⁶².”

We have added text to our discussion on lines 696-698 to make readers aware that some anthropic factors are not well captured in our study: *“For our study, we could not yet include rarely-available local variables that can more accurately recover species responses to environmental variation or anthropogenic threats⁷⁶. For example, we had to omit potentially important threats with no available data, such as multiple forms of pollution, the alteration of natural flow regimes through hydropower generation, and the location of extreme drought or thermal events.”*

Reviewer #2 (Remarks to the Author):

Thank you to the authors and editors for the opportunity to read this interesting manuscript. My review is below.

—Summary of Article—

The authors investigate the use of Shapley values to characterize the local drivers of habitat suitability and anthropogenic threat experienced by a species. They coin the term “shadow distribution” for the portions of a species’ (real or potential) range which could be suitable for the species were it not for the effect of anthropogenic threats. They explore this approach using a selection of 9 freshwater fish in Switzerland and a random forest SDM over subcatchments based on a suite of 18 potential covariates. This approach is used to dissect the differing species responses to environmental covariates, to inspect the spatial distribution of this variance, and the potential of a spatially explicit understanding of the drivers of species suitability to direct conservation decision making.

—Summary of Review—

As the authors claim, the application of XAI to SDMs is a natural extension of prior work on variable importance in SDMs and a potentially useful contribution to the effort to separate a species’ fundamental and realized niche. However, I believe more significant attention should be given in the main text to the definition and interpretation of Shapley values in order to draw a clearer line between their statistical and causal interpretations, particularly as they are related to conservation decision making and to the concept of a “shadow distribution”. That said, I am not aware of any existing literature that has presented these ideas and my technical concerns about the methods here are minor, so I recommend revisions to the paper need only be relatively minor.

We thank the reviewer for their constructive feedback and have made all the suggested changes, additions, and clarifications. We pay particular attention to the line between causal interpretations and statistical relationships in the methods and discussion.

—Significant Feedback—

- The article should be related better to existing variable importance work in SDMs. Specifically, how does a site-specific breakdown of variable importance from Shapley values compare to range-wide variable importance measures in random forests or other single-species SDMs? Ideally this would be explored in some of the case studies presented (i.e. comparing Shapley values w/ a species-wide measure), but additional context is necessary regardless in order to understand its relevance to conservation.

In summary, we have now better clarified that aggregated SHAP values that we already provided are well-accepted global variable importance indicators. We now show additionally that other global variable importance measures are highly correlated with SHAP based measures. We agree that global measures of variable importance are of high interest and provide a baseline to interpret the SHAP values with relevance to conservation. We provide further explanation below:

It is well accepted in the machine learning community that aggregated values of SHAP across all observations in the model are a global model interpretation method. Therefore, we did already provide a ‘range wide’ variable importance measure of random forests as requested by the reviewer. We performed new analyses presented below proving these are comparable approaches. These SHAP aggregations are well accepted and recommended as examples in text books XAI and SHAP (Molnar) and packages for SHAP in python (<https://github.com/shap/shap>; Lundberg and Lee 2017).

More specifically, a reader can find range-wide “global” variable importance as requested by the reviewer in:

- Figure 1(b) as the mean absolute value of the SHAP values
- Figure 1(c) showing the distribution of positive and negative SHAP values,
- The insets of Figure 1(e-l) showing the overall SHAP values across environmental gradients which indicates response curves of each variable on habitat suitability.

Further analysis supporting the above: To check the validity of SHAP as a global variable importance indicator, we compared two other measures of global variable importance frequently used for random forests and SDMs. We used the permutational variable importance which measures the loss of model performance (AUC) when randomising a feature. We also used the random forest specific approach of Gini importance which measures the total decrease in node impurity (i.e., improvement in split criteria) averaged over all trees in the ensemble. These measures of variable importance were strongly correlated with SHAP variable importance scores with a Pearson correlation of 0.88 ± 0.24 for Gini index and -0.84 ± 0.24 for permutational importance (negative due to being a loss). Given the low sample size these values and patterns correspond to a good match between methods (n per species is < 11, being the number of model features). We now show these relationships between SHAP performance and these other metrics in Figure S5 (see below) and reference them in line 253-255.

Figure S5. Comparison of SHAP global importance measures with traditional global variable importance measures. We compared SHAP global variable importance with two measures of global variable importance very frequently employed in analyses of random forests and SDMs. We used the model agnostic approach of permutational variable importance, and a random forest specific approach of Gini importance. Permutational variable importance measures the loss of model performance (AUC) when randomising a feature. Gini importance measures the total decrease in node impurity (i.e., improvement in split criteria) averaged over all trees in the ensemble. These measures of variable importance were strongly correlated with SHAP variable importance scores with a Pearson correlation of 0.88 ± 0.24 for Gini index and -0.84 ± 0.24 for permutational importance (negative due to being a loss).

After carefully considering the reviewers comments, we decided to keep the global model interpretation provided by the well-accepted method of aggregating SHAP values to provide a more

internally consistent set of analyses in the main manuscript. To address their concern, we better highlight that aggregated SHAP values are range-wide importance scores, and added to the methods lines 256-265.

“In addition to local interpretations, aggregating SHAP values across all observations in a model gives an indication of “global” variable importance... Note that overall importance of variables in determining species range wide distributions were comparable to traditional measures of “global” variable importance such as permutational variable importance scores (Figure S5).”

Lundberg, S.M. & Lee, S.-I. (2017). A Unified Approach to Interpreting Model Predictions. In: Advances in Neural Information Processing Systems. Curran Associates, Inc.

Molnar, C. (2022). Interpretable Machine Learning: A guide for making black box models interpretable. 2nd edition.

Molnar, C. (n.d.). Interpreting Machine Learning Models With SHAP.

- Some additional background on the definition and interpretation of Shapley values should be added to the main text. I appreciated and support the authors’ more technical presentation of Shapley values in Appendix 3, and found the included RMD vignette interesting, but found what was kept in the main text to do an insufficient job, especially for an audience potentially unused to variable importance in general and likely unfamiliar with XAI or locally-interpretable attribution measures. What is presented in the main text is of potentially limited utility to an unfamiliar reader (ex. what do the 4 game-theoretic optimality conditions of Shapley values have to do with interpreting environmental suitability/threats?). I suggest adding a conceptual figure (perhaps based on some of the material in Appendix 4?) and expanding the section in the main text so that a reader who reads only the main text will have a serviceable understanding of XAI and Shapley values.

We now include a conceptual figure (Figure 1). This figure helps the reader move from modelling spatial distribution of species occurrences in relation to environmental values, to the conceptual understanding of expected and shadow distributions, and how these are derived using SHAP values.

Further, we now better introduce SHAP in the introduction (main text) with the following on lines 62-65:

“Recent work shows that the spatial contribution of different environmental factors can be identified by applying explainable artificial intelligence (XAI) to SDMs 21,22, which highlights new avenues for generating fundamental and applied insights on the geography of environmental constraints on species populations.”

And lines 72-79:

“We apply an explainable artificial intelligence (XAI) using SHAP analysis combined with species distribution models to estimate local relative contributions of natural factors and anthropogenic threats to local predictions of environmental suitability. SHAP enabled us to decompose net habitat suitability score for each location into separate contributions from each environmental variable. We aggregated SHAP in a novel way to ask, what is the spatial distribution of positive effects of natural factors (i.e., a species abiotic niche) and negative threat effects on a species distribution?”

We rephrase the written text in the methods paragraph starting line 240 which now reads:

“SHAP values are an explainable artificial intelligence (XAI) tool to explain a prediction made by a model. SHAP values provide an interpretation of covariate effect on the predicted outcome at the observation-level in the model. A SHAP value indicates the difference between what a variable contributes to a prediction in each location, and what the variable is expected to contribute given the mean model prediction. Other variable importance approaches generally provide ‘global’ insight to variables importance across all observations in the model (e.g., permutational variable importance).

In contrast, SHAP provides a single value per observation per variable. This SHAP value indicates the featured contribution to the prediction for that specific data point. In a spatial model, the observation level is inherently linked to locations. In our models of species occurrence, a positive SHAP value indicates a given variable is contributing positively to the occurrence prediction (increases the prediction), and vice versa, and if it is 0 it has no contribution. We can compare SHAP values of all other variables in the focal location to understand relative importance of individual variables within a species distributions. Or, for the same site we can compare between species the relative contribution of different variables. SHAP values are model agnostic and so can generalise to any statistical model that explains variation in ecological properties across environmental gradients (e.g., abundance, biomass, growth rates, body condition, productivity, species richness).”

We hope these substantial additions now provide a good basic understanding of SHAP and its application in our work.

- Several of the manipulations of Shapley values are new to me and feel worryingly unusual. Specifically, the correlations of Shapley values w/ covariates, stacking of Shapley values to define a “shadow” distribution, and the hypothetical exploration of the effect of “mitigating” threats by modifying features according to their Shapley values. I suggest these concepts could be more well supported in the main text or appendices, as they get away from traditional uses of Shapley values in ways that are potentially unique and powerful.

In the below sections we answer the reviewers points in turn:

Correlation of SHAP values with covariates

Our manipulations of SHAP are not new in the context of machine learning. The correlation and relationships between SHAP values are a suggested way to interpret the direction and shape of covariate response curves as emphasised in Molnar “Interpretable Machine Learning with SHAP” (Page 87) as well as in environmental science publications using SHAP (e.g., Cha et al., 2021; He et al. 2022; Wadoux and Molnar 2022).

Stacking of SHAP

Our stacking of SHAP values to define, in part, the “shadow distribution” is valid because SHAP values follow the principle of “Efficiency” as show by Štrumbelj and Kononenko (2010), Štrumbelj and Kononenko (2014), and Lundberg and Lee (2017b). This means that the sum of the SHAP values + the mean prediction adds up to the local prediction of habitat suitability. Further, because feature order is irrelevant for SHAP (i.e., Symmetry axiom), and features not affecting predictions get values of 0 (Dummy axiom), this means we can interpret each features contribution separately. Together, these axioms mean we can sum subsets of covariate SHAP values and interpret the summed values within the relevant context of our study. As such, summing over subsets of features (threats) indicates the contribution of these features combined to the habitat suitability prediction. These summed values we used to define the expected distribution (line 295 onward) and the shadow distribution (line 333 onwards).

This logic is supported by the authors of foundational papers on SHAP (Scott M. Lundberg) in the package issue on github highlighted here <https://github.com/shap/shap/issues/933>.

We now add the short explanation to our methods section on lines 257-261: “*Due to SHAP values satisfying the efficiency criteria of interpretable XAI methods⁵² (summing to the predicted mean), summing subsets of variables by groups (e.g., summing across all threats) indicates contributions of groups of variables to the mean prediction. We calculated the mean absolute SHAP value which indicates a variable’s overall importance in changing model predictions.*”

Hypothetical exploration of the effect of “mitigating” threats by modifying features according to their Shapley values:

We have further clarified in paragraph starting line 362 that our approach to calculate shadow distributions depends on hypothetical scenarios which come with uncertainty. Therefore, we provide scenarios to explore this uncertainty within the SHAP based approach to estimating shadow distributions. We also now provide a second approach to estimate shadow distributions. We show in Figure S6 that the correspondence between different methods, and different assumptions, is very high (median Pearson correlation between approaches > 0.85). We keep the SHAP based analysis in the main text to ensure the manuscript is internally consistent and more easily comprehensible. In addition, our SHAP based approach enables that our quantitative estimate of the shadow distribution is methodologically comparable to the qualitative shadow distribution based on highlighting areas with negative SHAP values. To explain how we explored uncertainty in shadow distribution estimation, we added the following to the methods section of our manuscript lines 308:

“Our estimation of shadow distributions by adjusting SHAP values (e.g., $Q_{0.95}$ (SHAP)) is a hypothetical scenario and comes with assumptions and uncertainty. To understand the impact of these choices on our results, we generated two other hypothetical threat alleviation scenarios. We calculated a very conservative scenario by converting negative SHAP values to 0, in this scenario threats no longer have a negative contribution to environmental suitability (but the underlying factor also does not contribute positively to environmental suitability). Second, we converted negative SHAP values to the mean positive SHAP values for each threat factor, which indicates a positive recovery of threats to the average condition in unthreatened regions for each threat factor. In addition to our SHAP adjustment, we tested an approach to estimate shadow distributions where we adjust the feature values in the environmental data directly and compare the observed and expected distribution of suitability scores (see Figure S6). This approach simulates improvements in environmental states and makes new predictions given these improvements. In this approach, we replaced environmental values of threat features to be the 99th quantile if a high value represents an improved state (such as higher connectivity) or 1st quantile in the inverse case, such as lower morphological modification. We found the output from these non-SHAP method to be very highly correlated to the SHAP method presented in the main manuscript for estimating shadow distributions (median correlation across species = 0.88, IQR = 0.85-0.89; Figure S6). For consistency, we present here only the first described SHAP based shadow distributions described in equation 4, but note that shadow distributions, like geographic ranges, are latent properties, so perfect calculation is impossible and estimation methods are required.”

Cha, Y., Shin, J., Go, B., Lee, D.S., Kim, Y., Kim, T. and Park, Y.S., 2021. An interpretable machine learning method for supporting ecosystem management: Application to species distribution models of freshwater macroinvertebrates. *Journal of Environmental Management*, 291, p.112719.

He, B., Zhao, Y. and Mao, W., 2022. Explainable artificial intelligence reveals environmental constraints in seagrass distribution. *Ecological Indicators*, 144, p.109523.

Wadoux, A.M.J.-C. & Molnar, C. (2022). Beyond prediction: methods for interpreting complex models of soil variation. *Geoderma*, 422, 115953.

- An additional note should be added to the cautions and limitations section on the causal misinterpretation of Shapley values. I believe this is especially important given the experiments varying features to more “ideal” values wrt their Shapley values and the relation to conservation planning. Of course, Shapley values can provide useful insights to planners and managers, but readers should probably not be given the impression that Shapley values give us levers to control directly and quantitatively in habitat restoration efforts.

We now added to our discussion a paragraph on the interpretation of SHAP with regards to causality that also links to the comments and requests from reviewer 3, in the paragraph starting line 665.

—Minor Feedback—

- Much of the details on the reduction from 18 covariates to 11 could be moved to an appendix, especially to keep focus on the novel variable importance methods

We now shorten this paragraph to only retain the key steps. We moved the original longer description to the relevant section of the ODMAP protocol in Appendix 1.

- Similarly, the coupling/decoupling and convergence/divergence analyses were interesting, but could also move to an appendix.

We have removed the decoupling and coupling analysis in Figure 2, but kept Figure 3 due to positive reviews below from reviewer 3. We removed a paragraph of text from the results that focused on decoupling-coupling as suggested. When describing the effect of variables on species environmental suitability we now highlight that Figure 3 provides an overview of response magnitude and direction across all species:

*“The remaining factors of urbanisation, river morphological modification index, distance to lakes, floodplain availability and flow velocity had lower contributions (an overview of the magnitude and direction of all variable effects across species is shown in Figure 3). Investigating the spatial distribution of SHAP values revealed independent contributions of variables to the spatial distribution of environmental suitability of *A. bipunctatus* (Figure 1e-l, and across all species Figure 3, Figure S-7-S15).*

—Line Items—

L38-44: these lines don't scan well to me. Much of the intro could use some refocusing (e.g. also L72-77). We have restructured the introduction which now removes these lines.

L172-176: confusingly worded. Rephrased for clarity.

L185: be more specific about what “successfully removed” means (or cut some of this material, as suggested above). We have cut much of this material as suggested.

L204: exAI → XAI. Changed.

L272-276: doesn't scan. We have modified this section for clarity.

L304: what are “connectivity threats”? Rephrased to “We also averaged the presence of negative influences for habitat quality (defined above) and connectivity loss as anthropic threats”

L408: missing “of” in “14% sub-catchments”. Changed

L602: typo: “Previous largely” Rephrased to “Previous work often”

L606: typo: threat → threats Changed.

Reviewer #3 (Remarks to the Author):

The authors present a study on (anthropogenic) threat mapping of freshwater species and disentangling from natural environmental factors, resorting to correlative distribution, resp. habitat suitability modelling, followed by explainable AI (XAI) methods via Shapley additive values (SHAP) to identify contributing factors. The study overall appears diligently laid out, well thought through, and well executed. The methodology itself is not new, nor is it entirely solid (see comments below, especially regarding potential variance inflation). However, this is one of the first papers applying it in a fully proper way, with attention to resolving unwanted artefacts like collinearity. The key contributions of the work are twofold, including (i.) an introduction of a new concept ("shadow distribution"), and (ii.) end-to-end application and case study on well-monitored ecosystem and species.

Thus, this work certainly is of great value and worth publishing, upon having answered a few concerns I have. Please see major and minor comments below.

Thanks to the reviewer for their very thorough review, we have addressed all the below points in turn and think it has substantially improved the manuscript.

MAJOR COMMENTS

- The Introduction section is cumbersome to read and not entirely insightful. It feels as if the authors tried to fully separate methodology from concepts, but have gone a bit too far—for example, the present work basically builds on species distribution modelling (SDM), but that large branch of research is not even mentioned until later.

We have restructured our introduction to provide more focus on the core concepts presented. We have removed lines 60-68 from the previous manuscript version, these lines introduced the concepts of species-specific variation in coupling-decoupling which is no longer a strong focus of our manuscript. We now introduce SDMs in lines 56-59 and 62-65.

The inclusion of the Kunming framework is not done sufficiently—any study on environmental drivers could help the framework, but this work here can do so in a different way by looking at local effects. There are strongly supporting passages in the framework regarding this topic, which are currently unaddressed.

We have modified our introduction to better highlight where the GBF framework is relevant for our work. We were not sure exactly where in the framework the reviewer is referring to, but re-read the framework with specific focus on targets 1-3.

We especially highlight target 2 related to 30% of areas under restoration activity by 2030 on lines 49-53: *“For instance, the Target 2 of the GBF calls for 30% of degraded areas to be restored by 2030 but a gap exists in how to implement such goals, because we often do not where and which threats most strongly negatively impact species locally.”*

We also now state in the introduction of shadow distributions the relevance for the GBF in lines 87-92: *“ We then examined the extent that shadow distributions mask areas of potentially suitable habitat, which would indicate that using environmental suitability predictions from SDMs greatly underestimates the expected, or potential, distribution of species. If using indicators of species distributions from environmental suitability predictions alone (e.g., 29,30 and as indicators for GBF targets 1-3), such differences between raw environmental suitability and expected distributions could undermine the monitoring, implementation, and priority setting of any spatial biodiversity assessment.”*

- I am missing a critical reflection on the limitations of the methodological approach itself. While the section on "Cautions and limitations" (p.20f, l.609ff) talks about pitfalls in applying the methodology, there is nothing about its inherent shortcomings.

These include (among others) the following:

- a. Risk of variance inflation (p.7, l.240ff): It conceptually makes sense to relate covariates values to their local importance scores. However, you are doing this on (i.) Shapley values, which are a statistically uncertain model (and even worse so as you have to approximate), (ii.) RF predictions (yet another statistical model), and (iii.) residuals instead of measured values for some covariates emerging from a GAM (yet another one). You do so using a Spearman correlation (another statistical model). This sounds like a recipe for trouble due to variance inflation. My take is that while you certainly can do that, results should be interpreted with utmost care, as the likelihood of encountering a measurement that is too uncertain to be trusted, or worse, spurious correlation artefacts, is very high.

We now include this point in the discussion on line 677-691. We also added in the supporting materials Appendix 2 a new Figure S3 (not shown here) which indicates the very low correlation between all variables in our model, especially in the dimensions that are important for our definitions of expected distributions and shadow distributions.

Our understanding of the issue of variance inflation is that one could be interpreting spurious effects due to increasing errors of predictions, or parameter estimates, due to multicollinearity of regressor variables when using statistical and ML frameworks. To test this idea, we refitted our models using generalized linear models and multiple-regressions, to examine if our results from random forests were strongly impacted by multi-collinearity. Under strong multi-collinearity we would expect different parameter estimates, and highly inflated standard errors, when using a GLM vs. when using a random forest. However, we recovered identical and very convincing effects for all the major drivers of species distributions recovered in our random forests. We did not include the residual-corrected variables here because multiple regressions would essentially "double correct" these variables, and they had much weaker variable importances in our random forests. As such, the major findings of our variable effects are reproduced in a simple approach that would reveal if the main results from our random forests were strongly influenced by variance inflation, by showing higher standard errors and different effects to the random forests, if multi-collinearity and variance inflation were the true drivers of our random forests.

Review figure 1. Coefficient values and standard errors output from GLMs fitted using the same response data and covariates as presented in the main manuscript. These values indicate the same magnitude and direction of covariate effects using an approach that is inherently robust to variance inflation (which, if present, would greatly inflate standard errors, change the direction of covariate effects).

b. Shapley values require input covariates to be uncorrelated. The authors address this in a good way by de-correlating them via a GAM and regression splines and using only residuals for some variables (although they do so for facilitated interpretability reasons and never mention this requirement). It is obvious that this introduces yet another level of uncertainty (model choice and parameterisation), and that this is not the only possible solution to this end.

We followed Review 2’s advice and moved much of the explanation how we decorrelate variables to the supporting information. However, we now add to the methods on line 204-208 “A fundamental aim of our work is providing interpretable (understanding inner workings) and explainable (understanding why a prediction is made) models. Multi-collinearity induces challenges in interpreting the independence of variable effects and interpretation of SHAP values 40. Through the below procedure our final variables were highly decoupled having a median absolute correlation of 0.05, a 95th quantile of 0.26 (Figure S3).”. We add to the discussion multiple points addressing causality, multi-collinearity, and interpretability starting lines 677.

c. Shapley values themselves are not the only means for XAI. The authors briefly mention one competing method (LIME) and cite other papers on why their method is to be preferred. This by itself is not wrong; Shapley values have shown to be very robust and less susceptible to unwelcome error sources indeed; it is also okay to not elaborate too much on the technical terms here, as this is not the key focus of the paper. However, this likewise has to be mentioned as a potential error source.

To acknowledge that the use of other XAI tools could modify the insights gained we now include the following on lines 703: “Further, we chose SHAP as our XAI tool which is an additional source of unexplored uncertainty, many other tools exist with different mathematical axioms some of which may provide alternative insights (see ^{53,84}).”

- The Conclusion section is detached from the antecedent text and underwhelming. Please pick up again the gist/major messages from the paper.

We now rewrite our conclusions to focus more on the main concept in our paper: the deconstruction of species spatial distributions, and the use of expected and shadow distributions in biodiversity-threat assessments.

Our conclusion now reads:

“For biodiversity conservation, protection and recovery we must identify and contextualise threat impacts within the multiple natural constraints on species distributions. We show how to identify when threat impacts occur in portions of species geographic distributions that are naturally highly suitable. We highlight an important decoupling between the different factors that determine species distributions. We define species’ expected distribution and species’ shadow distribution to help quantify the magnitude of this decoupling. Our work suggests indicators for national Biodiversity Action Plans underlying the Kunming-Montreal GBF based on species distribution models should also consider expected and shadow distributions. Failing to do so, we miss insights to the negative influence of anthropogenic threats on species distributions. Our work supports the assessment of threats to biodiversity at large-scales, and moves towards a framework tailoring conservation actions to local threats demonstrated to impact species distributions.”

MINOR COMMENTS

- p.1, l.27: "widely used threat and biodiversity mapping exercises" → too vague

We have now changed this sentence to be more specific and read on lines 27 *“Our findings highlight conservation of species geographic distributions is likely insufficient when biodiversity mapping is based on species distribution models or threat mapping without also quantifying species’ expected or shadow distributions.”*

- p.2, l.35ff: Would it be possible to add an example on such "specific threats" (i.e., elaborate on why a "simple" expansion of protected areas does not always suffice)?

In updating the manuscript based on previous comments about the introduction, we no longer make a strong contrast between “actions” and protected areas, so have not addressed this comment which we think might be less relevant now.

- p.2, l.44ff: A mention of the Kunming GBF is of course almost a given here, but I would recommend providing a bit more detail (e.g., highlight some of the goals of the framework that specify and rely on local conservation actions). Otherwise, this could just as well be mentioned for global studies (it currently does not add much to the "local" argument).

Comment addressed in previous responses above.

- p.2, l.47: Consider replacing "downscaling"; this has an unjust negative connotation here.

Removed as suggested.

- p.2, l.60f: Rather strange sentence ("when mapping and ranking threats, researchers assume that threats are important (...)"). Consider reformulation.

This sentence has been removed when restructuring the introduction.

- p.4, l.126f: what kind of "modern threats" are potentially absent from data of 2010 and earlier? In other words, can you explain in more detail why you limited measures to 2011 and newer?

We have changed the sentence to now read as follows, which we hope is clearer. If we included older records (e.g., from 1950s) we might overlook that threats were implemented, or intensified, in the more recent era, so include records of unthreatened populations.

"We performed our analyses on data collected after 2010 to avoid potentially including records that indicate species presence before the masking the effect of modern threats by have impacted species' populations causing local extirpation."

- p.4, l.131f: Did you aggregate observations in time? Would it make sense to conduct analyses with time as a factor? I could imagine threats having changed over time, for example due to the Covid-19 pandemic.

This is interesting, but we believe beyond the scope of this work. We aim in the future to perform time-series analysis on data we are currently compiling but are not available yet for full analysis.

- p.5, l.149: Did you mean "local values *within* 100m from river"?

Changed

- p.5, l.157: What do you mean by "downsampled" random forests?

We now highlight this is in the sense of Valavi et al. 2021. And explain on lines 195 *"In this down-sampling procedure, each tree is fitted to a balanced sub-sample of presences and absences."*

- p.5, l.159f: "(...) and perform as well as model ensembles" → Sort of illogical clause. The performance of RF heavily depends on the experimental setup and may not always be better than an ensemble of models; also, strictly speaking, RF already is an ensemble (of bootstrapped decision trees). Consider dropping clause.

We think it is important that readers know that down-sampled random forests have been shown to perform very well in comparison with other very common species distribution modelling approaches, whereas non-downsampled random forests can strongly overfit leading to low performance. We rephrase to on lines 190 to *"Random forests perform well at prediction tasks across multiple data types, and have been demonstrated to perform as well as model ensembles in modelling species distributions^{40,41}."*

- p.5, l.161f: Class imbalance does not lead to overfitting, but to biased predictions. The concept of overfitting pertains to performance issues of a converged model when deployed on out-of-training set data. Also, downsampling is a heavily compromised approach to counteract imbalances as it drops potentially vital training points; did you consider other remedies?

We employ the method highlighted in Valavi et al 2021. Down-sampling does not remove data points in the entire forest, but down-samples each individual tree to be balanced. Therefore, all the data is used to fit the model but each tree has a balance of presences and absences. Valavi state that class imbalance leads to overfitting training data. We now rephrase on lines 193 to *"We used down-sampling to address the class-imbalances that can lead to model overfitting training data"*

- p.5, l.163: what was the "balanced subsample" size (how many presence/absence points per decision tree)?

We state in lines 199 *"We set the ntree to 1000, the down-sampled "sample size" to be the minimum of either class (0 or 1) and used the default mtry parameter (the square root of the number of*

covariates).” We refer readers to Valavi in the first line of random forest descriptions to see this exact implementation.

- p.5, l.163ff: Strictly speaking, RF also do not provide occurrence probabilities in unmodified settings; all they report is the fraction of agreement across the trees. No action needed; I approve of the term "environmental suitability".

- p.5, l.167f: Did you conduct a grid search, resp. hyperparameter tuning stage, or are these parameters just set by hand? Also, please refrain from referring to "default" parameter values, as a default is generally implementation-specific.

We did not perform a grid search or model as RFs are known to work reasonably well with the parameters we used and our models show adequate performance for interpretations. We have changed the sentence on lines 201 to read: “...and set the *mtry* parameter to the square root of the number of *covariates*. We follow³⁹ in not further tuning random forests parameters which exhibit low tuneability^{42,43}”.

- p.5, l.169: You are addressing the elephant in the room here with multicollinearity. This is not (just) because of the SDM (RF can work very well even if covariates are collinear) and importance interpretation, but because of the Shapley values framework itself you use for XAI. A vital requirement for these is covariate independence, a property often ignored by existing studies. I strongly recommend mentioning this requirement (with a reference) at an appropriate stage in the manuscript.

We now include the following on line 204 before introducing how we considered multi-collinearity: “A fundamental aim of our work is interpretability (understanding inner workings) and explainability (understanding why a prediction is made) of our models, and collinearity induces challenges in interpreting the independence of variable effects and interpretation of SHAP values⁴²”

- p.5, l.176ff: I really like the idea of using GAM residuals to test for confounders. Since RF can model nonlinear relationships, it makes sense to test for nonlinear collinearity effects as well here. I assume this is why you used regression splines. I am wondering though how much effect a different choice of data transformation and/or link function in the GAM could have on the outcome of the collinearity analysis. Did you try other nonlinear variants to see what the outcome is?

We did not try multiple types of transformations and link functions but used that which was most appropriate for each variable being controlled. For example, cropland cover, varying between 0-1 was modelled with a beta error distribution and a logit link function setting family = betar(link = 'logit'). We note in the discussion on lines 677 that the choice of model to remove multi-collinearity is an additional source of variability as in comments above (a-c) from reviewer 3.

Aside, if you mention the software used (mgcv), please do so as well for random forest.

We have added randomForest version.

- p.5, l.179ff: Again, using residuals as input is an intriguing idea. It does sound logical, but its accuracy all (again) hinges on the quality of the GAM used. This could benefit from additional analyses.

We include in the discussion lines 682 that this could be a further line of research to evaluate optimal solutions to this problem (as this was not the main aim of our manuscript). We note that two-step / sequential regressions have a long history in ecology in dealing with multicollinearity (e.g., Graham et al. 2003), and also highlight that modern advances in applied statistics for observational datasets are

still grappling with this complex problem and converge on a similar (but linearized) version of our approach (e.g., Feng and Chen 2022).

Graham, M.H. (2003). Confronting Multicollinearity in Ecological Multiple Regression. *Ecology*, 84, 2809–2815.

Feng C, Chen X. A two-stage latent factor regression method to model the common and unique effects of multiple highly correlated exposure variables. *J Appl Stat.* 2022 Oct 25;51(1):168-192. doi: 10.1080/02664763.2022.2138838. PMID: 38179159; PMCID: PMC10763915.

- p.6, l.200: What do you mean by "final random forests"? Did you use the five-fold CV for hyperparameter tuning (e.g., no. trees, which covariates to use) and had a separate, held-out test set, or for testing of the final model? → Unclear.

We now remove this sentence, and add to the beginning of this paragraph the below following text on lines 222. We fitted models for cross validation to assess out-of-sample model performance, and then used random forests fitted to all the data in the main manuscripts.

“Random forests presented in the main text were fitted to all available data for each species, but first we assessed model performance using...”

- p.6, l.202ff: I would mention the term "local feature importance" here explicitly, as it is common and has been around for quite some time in the machine learning community.

Added term as requested.

- p.6, l.204: "XAI" is a more common acronym for the concept than "exAI" I believe.

We changed all instances of exAI to XAI.

- p.6, l.214f: This statement simply is not true. There are many more references beside the three mentioned ones, such as:

Cha, Y., Shin, J., Go, B., Lee, D.S., Kim, Y., Kim, T. and Park, Y.S., 2021. An interpretable machine learning method for supporting ecosystem management: Application to species distribution models of freshwater macroinvertebrates. *Journal of Environmental Management*, 291, p.112719.

Bourhis, Y., Bell, J.R., Shortall, C.R., Kunin, W.E. and Milne, A.E., 2023. Explainable neural networks for trait-based multispecies distribution modelling—A case study with butterflies and moths. *Methods in Ecology and Evolution*.

He, B., Zhao, Y. and Mao, W., 2022. Explainable artificial intelligence reveals environmental constraints in seagrass distribution. *Ecological Indicators*, 144, p.109523.

Miyaji, R.O., Almeida, F.V. and Corrêa, P.L.P., 2023, September. Evaluating the Explainability of Machine Learning Classifiers: A case study of Species Distribution Modeling in the Amazon. In *Anais do XI Symposium on Knowledge Discovery, Mining and Learning* (pp. 49-56). SBC.

Ryo, M. and Angelov, B., High accuracy is not enough: Interpretability in species distribution model increases with explainable artificial intelligence/interpretable machine learning.

Thanks for pointing us to this literature, which we have now read and included in our work. The literature base is still very new (mostly < 5 years old) so we rephrase this line on 236 to read:

“The application of XAI in ecology and conservation is nascent^{34,48} with the few implementations of XAI with ecological models showing great potential to generate novel insights into complex ecological phenomena^{34,49–55}.”

- p.7, l.258ff: I understand the motivation behind this, but what do you do if a variable is an abiotic, contributing factor and *can* be directly influenced by human actions?

This is an assumption of our approach that we clearly highlight in the methods. This point was also raised by reviewer 1 (see last comment of reviewers 1's responses) and is now clarified in the methods stating on lines 297-303.

- p.7, l.259ff: This in turn I am not sure about. Are Shapley values really comparable across covariates? My intuition tells me that this is true if the covariates are normalised (e.g., by Z-scoring). However, you are also feeding residuals into the model. This merits further investigation. The same goes for the threat alleviation setup following right after.

SHAP values are on the scale of the output (habitat suitability) so are a comparable unit across features in the model. To double check this concern, we have refitted an example randomForest and plotted the SHAP values against the SHAP values of a random forest with standardized (0-1) feature values. The correlation is 0.994. Given that both random forests, and SHAP estimates, are stochastic processes, this is therefore an almost perfect correspondence between standardized and non-standardized feature values. This is also supported in the user manual of SHAP (https://shap.readthedocs.io/en/latest/example_notebooks/tabular_examples/linear_models/Explaining%20a%20model%20that%20uses%20standardized%20features.html).

The above figure shows the SHAP values when a random forest for *Alburnoides bipunctatus* is fitted and SHAP analysis performed on standardized vs. unstandardized variables.

- p.8, l.296ff: I've been expecting the shadow distribution to be defined as the difference between expected and observed. This intuitively makes sense—but only if the expected distribution can factor out *all* disturbing factors. This relates to the discrepancy between what is observed regarding species (presence/absence) and what is measured with respect to environmental factors; in a nutshell, it boils down to the problem that observations exhibit the realised, and measurements the fundamental niche. This is an age-old conundrum in SDM works. Now, my key concern lies in the fact that the realised niche is not just influenced by abiotic drivers and human disturbances (as modelled by you), but other, non-considered factors as well, including any type of biotic interaction. Hence, even though you make a good effort in defining disturbance factors, your "shadow distribution" is nonetheless confounded by ignored factors like biotic ones, and should be interpreted with care.

As the reviewer highlights, this is a longstanding and fundamental issue with all applications of SDMs from observed data. We now better acknowledge this limitation in lines 706 with specific reference to shadow distributions: *“Caveats applicable to all SDM models also apply to the interpretation of shadow distributions, for example, whether the model of the realized niche accurately represents the species fundamental niche influences how well the deconstructed environmental contributions reveal the shadow and expected distributions. Future work could better reveal how species interactions influence species distributions and identify where a lack of key interactions shift the location of expected, and therefore shadow, distributions”*.

- p.10, Figure 1: Multiple issues:

- Panel (b): Could you add whiskers to the bar chart indicating e.g. the standard deviation of each average importance?

- The individual graphs of the subpanels are rather hard to read.

- Resolution is low. Do you have a better quality version of it, ideally with vectorised textual descriptions?

- p.10, Figure 1: Moreover, I believe parts of this figure to exemplify the limitations I see with the approach. Let us consider discharge—unsurprisingly, values are highest directly measured over major rivers (panel (e)). Just as logically, occurrence of the species is highest in rivers (panel (a)). Unsurprisingly, river discharge then emerges as the most important contributing factor (panels (b), (c)). However, the vast majority of this explanatory effect just states the obvious (i.e., fish don't naturally occur outside water bodies). Worse, it may masquerade bias effects you may not have considered—for example, RF may predict highest suitability in the biggest rivers (Aare, etc.), simply because the species was observed more there. Accordingly, discharge again emerges as the biggest contributing factor, because these rivers are large. What if a certain species actually finds better conditions in smaller tributary rivers, but this just does not emerge because of sampling biases? → In sum, XAI methods suffer from precisely the same biases and issues as prediction models (SDMs) in the first place.

The issue of sampling biases highlighted here is only valid for non-probabilistic samples, namely SDMs build from presence-only data. The raised issue is therefore not present in our models because we only used presence-absence data from standardized surveys that record all species present (and absent/not detected) in a location. In fact, there are fewer surveys in the largest rivers due to challenges with surveying large water bodies with the electrofishing technique. The general issue that sampling biases confound presence-only SDMs, and also, therefore SHAP explanations of presence-only SDMs is now highlighted as a warning in our limitations on lines 711 (but is actually not applicable to our analysis):

“Further, any issues relevant for presence-only SDMs, such as sampling biases, are also problematic for SHAP explanations of these models, and as such we encourage the use of presence-absence data from standardized surveys to build SDMs.”

- p.11, l.360ff: Indeed; this is the root of it all. Do you by any chance have examples from the literature on how your focal species react to threats? If another study found that e.g. *Gobio gobio* was more sensitive to urbanisation than *Cottus gobio*, for example, and you could replicate this hypothesis with your approach, it would greatly add value to your work!

- p.11, l.368f: This is just as interesting and could lead to lots of work regarding e.g. invasive species.

- p.12, Figure 2: Great figure! I initially was a bit skeptical about comparing species even though they were fitted using independent models, but since the sampling design is the same and biotic interactions are explicitly left out, this should not cause problems.

I believe the side captions on the right for (iii.) and (iv.) are reversed, compared to the figure caption.

The above three comments conflict with the suggestions from reviewer 2. We have now removed figure 2 and the lines in the above comments as we and reviewer 2 felt it would aid the flow of the manuscript. So we do not respond to the above points.

- p.13, Figure 3: This figure is hard to read in monochrome/red-green blindness. Can you use a better colour ramp, or even better, also add different shapes to the points? Also, maybe highlight *O. mykiss* explicitly since it's the non-native one. Speaking of which, I would love to see some discussion on this species' response to its outlier positions in particular, such as flow velocity and morphological modification index.

We have modified the figure colour scale using this online tool (<https://davidmathlogic.com/colorblind/#%23D81B60-%230E0084-%23FFC107-%23085246-%23f46b00-%23c48cfe-%237f3ba7-%2390d27d>) to better assist creating a palette that allows differentiation for colour blind readers. The palette we now use has the RGB values #D81B60, #0E0084, #FFC107, #085246, #F46B00, #C48CFE, #7F3BA7, #90D27D, #000000. Rainbow trout are heavily stocked in Switzerland, which we expect to drive the weak association with connectivity, and inverse relationship with factors related to habitat quality (e.g., stocking occurs mostly in areas of low ecomorphological complexity).

- p.15, Figure 4: Again, very interesting figure! One minor question of interest: you explained how you calculated habitat quality above in the text (as a combination of multiple factors). Here, I see poor habitat in some areas in the central-north part of the country, but also in many other places together with poor connectivity (panel (c)). Can you provide further information on what exactly it is that makes these regions unsuitable habitats?

In general, many of the rivers in Switzerland are highly modified having no natural floodplains, low morphological complexity, being highly channelized, straightened, with artificial sides and sometimes bottoms. We outline these as drivers of biodiversity loss and population reductions in modified rivers in the paragraph starting 313.

- p.18, l.505: I would be careful with weighting terms like "successful" in this context.

We removed the word successful.

- p.18, l.513ff: This second contribution isn't all that new; I give credit that you were the first to properly address it in an SDM context (with sensible heeding of statistical requirements, such as variable independence) and with separation of natural/nonnatural threats.

We now cite others with similar goals in this statement.

- p.19, l.545ff: Sure, but the term "shadow distribution" first needs to be accepted by the community. Consider down-toning statement and targeting the idea, rather than the term you coined yourselves.

We rephrased to emphasise the concept and the term on line 605: "*A research agenda on shadow distributions, i.e., areas where threats negatively impact species natural distributions...*".

- p.19, l.551f: You are crossing scales and mapping units by a lot here (from regional to global). While the application of your framework in global studies may work, it is not clear how trustworthy results become then, and how well drivers can be separated with respect to latitudinal gradients.

We intentionally want to highlight to readers to think beyond regional case-studies presented here and consider parallels with the environmental determinants of global species distributions, for which there are longstanding fundamental questions in macroecology. We agree, the assumptions in doing so should be considered case by case (we do not suggest to apply our exact modelling exercise at a global scale).

- p.20, l.596ff: I am not sure I fully see through your line of argumentation here. Those two postulated sources (species can respond differently to environmental gradients and vice versa) sound rather obvious and not really related to the study in full. Also, there's a linguistic problem here (facsimile: "two major sources of variation contributing to responses: i) species having different responses")—the source of variation in response is differences in response...

We agree this is rather a complex line of reasoning and have remove this from our manuscript.

- p.20, l.599f: Above you claimed the opposite (w.r.t. latitudinal gradient).

Our argument here is that a diversity of responses to environmental gradients are common, as referenced also in the introduction. On line 599, we argued the main environmental determinant of fitness/population dynamics at latitudinal range limits is often not well understood, but has been questioned for centuries. We see these as different and individually valid points that do not conflict. We therefore have, on this occasion, not made a change.

- p.20, l.602: Not really. Location-specific XAI methods have been around for a long time; I don't see this as your primary contribution. Rather, your study is the first to study it in-depth in an SDM context and the first to provide further-reaching analyses w.r.t. disturbances (as opposed to just environmental factors).

We rephrased to de-emphasise this point as novel on line 655:

"Whilst diversity of species responses to environmental gradients are well-recognised, our work can help reveal how observed local biodiversity, and biodiversity change, arises from independent responses of different species to environmental change in any specific location."

- p.20, l.609ff: While it is common practice to place a limitations section at the end, I strongly advise adding some limitations above, perhaps already when introducing the methods. As said in one of the major comments above, there are quite some more pitfalls regarding your methodology than you write about here.

See major comments.

- p.20, l.612ff: You do have a point here, but there is a circularity to the argument: precisely because many large and/or multi-species models are tuned for accuracy, XAI methods like Shapley values are often a first step towards actually understanding the response curves.

We mis-phrased our point and instead meant to focus that before calculating shadow distributions, models should be interpretable and well understood (of course there could be multiple ways to calculate the concept of the shadow distribution and here we used SHAP), lines 665 now read: *"We suggest a cautionary inferential approach where each species response curve is well understood and trusted before quantifying shadow distributions"*.

- p.20f, l.615ff: I don't like this as a general statement, as not all applications require explainability over high accuracy.

We do not claim all applications require explainability, but in our experience, the majority of applications of SDMs largely ignore interpretability and explainability over predictive accuracy. We the explanation of our argument to make this point clearer on lines 671:

“However, making decisions requires well-understood models in more local-to-regional contexts for fewer species, and if the stakes of environmental decision making are high and contain financial costs⁷⁷ then trusting predictions of black-box models^{34,49} is risky. In such cases, interpretability and explainability could be higher priority than predictive performance.”

Reviewer #3 (Remarks on code availability):

Disclaimer: I have not run the code (and thus ticked "No" on the code review question), but have read through it.

Code is at academic/scientific level (not enterprise-grade) but seems legible and intuitive. Supplemented walkthrough document (Appendix 4) is excellent and explains code step-by-step. No major blunders detected.

Thanks, we appreciate the time for this very thorough review.

REVIEWER COMMENTS

Reviewer #1 (Remarks to the Author):

I have reviewed the revised manuscript. I now understand the context and intention of introducing the new theoretical concept in combination with a practical SDM approach, thanks to an extensive improvement on the manuscript. I admit that I misunderstood the authors' motivation for this study, and now, I am convinced by the value of this study. It is certainly beyond a typical XAI application study, and indeed, it provides a useful, valid technique to estimate the human influences on species distribution and inversely infer the extent of realized niche when no human factor is affected. In particular, Figure 1 can help readers understand the value of this study and how a new theoretical idea can be practically tested with a conventional SDM approach. They also nicely elaborated on the pros and cons of this approach. Given an exceptional contribution to developing a novel practical method for species conservation, I support their study for publication in Nature Communications.

Reviewer #2 (Remarks to the Author):

Having read the authors' revised manuscript and their responses to feedback from myself and the other reviewers in round 1, I find the update to be a significant improvement to the original manuscript and generally ready for acceptance. I have high confidence in the impact of the paper and believe the authors' improvements to the manuscript have made a large difference in the interpretability and applicability of their method. Overall, I recommend it for acceptance with some minor suggestions below. I'd like to thank the authors for their thoughtful responses to my review and wish them luck with this line of research!

I have also reviewed the supplementary material and found it to be of good quality and accurately presented, but have not recorded line-by-line feedback.

—Suggestions—

L27: “Our findings highlight [that] conservation of species[’]”

L46: species → species’

L50: achieve → achieving

L51: the Target 2 → Target 2

L51-53: this argument about Target 2 and threat assessment is not complete to me and needs to be revised

L130: “local” is repeated

L191: “and” → “an”

L193-195: I don't think these parameter settings are critical and could move to the modeling

part of Appx. 1

L196-197: randomForest ought to be cited, along with other packages cited throughout the manuscript and appendices (e.g. mgcv and BORUTA)

L197: missing period

L198-L215: this ¶ is almost fully replicated in Appx 1 and could probably be summarized for concision in the main text

L227: “the ‘local feature importance’ [or] the ‘situational importance’ of”

L229: replace “which is known as...” with a normal acronym definition

L230-233: cut, move the pointer to the appendix elsewhere and any otherwise uncited XAI papers to L234

L234: consider adding a one-sentence summary of XAI in general here (or somewhere earlier)

L302, 318, 332, L344: I don't believe these equation definitions are completely well defined. The use of \in to define a set (i.e. “binary expected dist'n \in ”) is not correct, and the Σ notations index over N and A while the subscript on SHAP is always i . In L332, it's not clear to me whether n is being used as set intersection or a logical OR. I generally recommend against using n unless it is clear from context that the operands are both sets and n is the set intersection operator. As an example of the notation (set builder notation) I recommend, I would write L302 as “binary expected distribution := $\{s_i \mid \forall i \in M \mid \sum_{j \in S} \text{SHAP}_{\{ij\}} > 0\}$, where s_i is the i th subcatchment in the set of all subcatchments M and S is the set of all abiotic variables.”. These equations should also probably be numbered. Overall, I found this section (L289-371) to be far too wordy and recommend cutting it down significantly and focusing the text more on the interpretability of these dist'ns. My expectation is that the two halves (expected and shadow dist'ns) could be condensed into one section to avoid duplicated and wordy variable definitions and allow for cleaner storytelling.

L307-317, L338-344: is the expected distribution a set? a value calculated globally across a species' distribution? or a summary value at each site/subcatchment? This isn't clear at all from the text or equation.

L319: if you're writing equations out, I suggest writing the subequation for \hat{y} as well below the one in L318

L320: the \rightarrow this

L322-L325: I don't fully understand this and missed it entirely on my first readthrough

L327: species \rightarrow species'

L344: I don't believe $f(x_i)$ is defined anywhere? I think this is the missing equation from L322-325, but it really should be defined concretely if you're giving equations for these

L568: “theory, that” → “theory: that”

Thank you once again to the authors and editors for the opportunity to participate in the review of this manuscript. The new version is a significant improvement and I believe a very promising piece of work.

Reviewer #3 (Remarks to the Author):

I would like to thank the authors for addressing most of my comments and providing in-depth answers. Thank you in particular for the following modifications, explanations, and additional experiments:

- The rephrasing of the Introduction and inclusion of (i.) contextual motivations like the GBF, and (ii.) SDMs. Much, much better overall!
- The inclusion of considerations regarding collinearity.
- The experiment on unmodified vs. 0-1 standardised covariates.
- Explanations regarding the downsampling procedure of decision trees. There seems to have been a misunderstanding on my end; this is now clear (also applies to other areas regarding the discussion; last few minor comments on original version).

Comments on rebuttal:

"We have modified our introduction to better highlight where the GBF framework is relevant for our work. We were not sure exactly where in the framework the reviewer is referring to, but re-read the framework with specific focus on targets 1-3."

Correct, your amendments touch more or less on what I had in mind. The framework outlines a number of headline and component indicators, often on a species or larger spatial level (i.e., global to country). As is expected from such aggregated measures, they do not highlight local issues enough. Hence, the GBF also contains supporting indicators that are designed to pick up details at a finer scale, which is precisely what your study could latch on. For example, CBD/COP goal A ("integrity, connectivity, and resilience of all ecosystems") contains headline/component indicators like the red list or living planet index (aggregated measures), whereas complementary indicators go down to more granular themes like land cover, algae cover, etc. All of these have to be measured and understood at all levels. Since we will inevitably use correlative models, employing XAI methods is arguably the best way to go; since it all boils down to anthropogenic change, disentangling contributions by type and introducing concepts like the shadow distribution do so as well.

Remaining larger comments:

- "The issue of sampling biases highlighted here is only valid for non-probabilistic samples, namely SDMs build from presence-only data. The raised issue is therefore not present in our

models because we only used presence-absence data from standardized surveys that record all species present (and absent/not detected) in a location."

I fundamentally disagree. It is very well possible to have sampling bias-related issues under systematic presence-absence surveys also, just to a lesser and different degree. You mention yourselves that largest rivers have received fewer surveys; this doesn't mean that biases have been cancelled out. Paraphrased: one can easily set up systematic and rigorous presence/absence surveys but fail to distribute survey locations across habitats evenly. Hence, the following sentence:

"Further, any issues relevant for presence-only SDMs, such as sampling biases, are also problematic for SHAP explanations of these models, and as such we encourage the use of presence-absence data from standardized surveys to build SDMs."

- p.8, lines 902ff: "To calculate the SHAP values exactly is extremely computationally challenging because the number of possible coalitions increases exponentially with the number of variables, and predictions from all possible combinations of variables must be calculated with and without the focal feature." – this, in principle, is correct. However, you are using a tree-based model (random forest), and for those, a computationally tractable method that is exact, known as "TreeExplainer", has been presented (Lundberg, S.M., Erion, G., Chen, H., DeGrave, A., Prutkin, J.M., Nair, B., Katz, R., Himmelfarb, J., Bansal, N. and Lee, S.I., 2020. From local explanations to global understanding with explainable AI for trees. *Nature machine intelligence*, 2(1), pp.56-67).

In principle, this begs the question why you used an old, expensive, approximative Monte Carlo method. Sure, you present your method as being model-agnostic, but this by itself is a bit tricky (you cannot apply these methods reliably to models like e.g. convolutional neural networks). Realistically speaking, I do not expect major differences to emerge across these methods; also, not using that version, despite its objective advantages, is of course by no means a death knell to the paper. However, for a journal like this one, I cannot leave the topic unaddressed, especially since SHAP values play such a central role in the methodology and others may want to implement your proposition. As a minimum, TreeExplainer needs to be referenced and (with a few sentences) addressed. An even better outcome would be to have a comparison between the Monte Carlo approximation and TreeExplainer on a subset of the data. Do you think this would be possible?

Minor comments:

- p.3, l.544: You may want to define the "SHAP" acronym here (first mention).
- p.3, l.549ff: I recommend rephrasing; questions like that don't look well in a paper in my view. Also, not sure about the novelty claim regarding that point.
- p.4, Fig.1: nice new figure, but again of low resolution and problematic colour ramps/combinations. You will likely be asked by the journal to improve that (also was a comment in my original review that remained unanswered).

- p.7, l.712f: Thank you for adding this explanation, but you perhaps want to move it down not to disrupt the natural flow of information (CV first, then the final model).
- p.8, l.874ff: Again, just an ordering issue: you start by stating that you "provide a detailed explanation of SHAP values in Appendix 3", then follow it up with a (higher level) description right after. I would move that sentence about the supplementary explanations after line 891.
- p.11, l.1173-1191: this is neat idea, but it in particular requires discussing limitations and pitfalls of XAI methods (we're "playing" with SHAP values here in a simulated causal way). You may want to refer to this part again in the limitations section at the end of the discussion as an example of where to be cautious (cf. p.22, l.1893ff).
- p.19, l.1701ff: I like this part a lot. One aspect that would make it complete as an argument could be to add examples of how to measure human disturbances for other taxa, even if just in a parenthesis.
- p.22, l.1883ff: Thank you also for that passage; great as well – apart from one detail (lines 1885f): SHAP values are not only "not inherently robust to correlated features", they require special treatment unless features are fully independent (Lundberg, S.M. and Lee, S.I., 2017. A unified approach to interpreting model predictions. Advances in neural information processing systems, 30).
- p.22, l.1898f: It's more as you have put it in the beginning of the paper: XAI methods describe both feature importance and model decision in a confused way.
- p.23, l.1964ff: Thanks, this is better.

RESPONSE TO REVIEWERS' COMMENTS

Reviewer #1 (Remarks to the Author):

I have reviewed the revised manuscript. I now understand the context and intention of introducing the new theoretical concept in combination with a practical SDM approach, thanks to an extensive improvement on the manuscript. I admit that I misunderstood the authors' motivation for this study, and now, I am convinced by the value of this study. It is certainly beyond a typical XAI application study, and indeed, it provides a useful, valid technique to estimate the human influences on species distribution and inversely infer the extent of realized niche when no human factor is affected. In particular, Figure 1 can help readers understand the value of this study and how a new theoretical idea can be practically tested with a conventional SDM approach. They also nicely elaborated on the pros and cons of this approach. Given an exceptional contribution to developing a novel practical method for species conservation, I support their study for publication in Nature Communications.

Thanks for the positive comments and perspective of our work, and the effort in helping improve our work.

Reviewer #2 (Remarks to the Author):

Having read the authors' revised manuscript and their responses to feedback from myself and the other reviewers in round 1, I find the update to be a significant improvement to the original manuscript and generally ready for acceptance. I have high confidence in the impact of the paper and believe the authors' improvements to the manuscript have made a large difference in the interpretability and applicability of their method. Overall, I recommend it for acceptance with some minor suggestions below. I'd like to thank the authors for their thoughtful responses to my review and wish them luck with this line of research!

Thank you for your efforts in reviewing our manuscript and we are happy to hear you think it will be a valuable contribution.

I have also reviewed the supplementary material and found it to be of good quality and accurately presented, but have not recorded line-by-line feedback.

—Suggestions—

L27: "Our findings highlight [that] conservation of species[']"

Changed

L46: species → species'

Changed

L50: achieve → achieving

Changed

L51: the Target 2 → Target 2

Changed

L51-53: this argument about Target 2 and threat assessment is not complete to me and needs to be revised

We added in revised lines 51-54 the following, we hope this is clearer. *“For instance, Target 2 of the GBF calls for 30% of degraded areas to be restored by 2030. However, we often do not know where and which conservation actions will be most effective because the local contribution of each threat to the state of species populations is unknown - limiting capacity to implement and downscale biodiversity targets effectively.”*

L130: “local” is repeated

Removed the repeated local.

L191: “and” → “an”

Changed

L193-195: I don't think these parameter settings are critical and could move to the modeling part of Appx. 1

We kept this sentence provide sufficient detail for readers to critically evaluate our modelling choices without requiring reading the details of the appendix.

L196-197: randomForest ought to be cited, along with other packages cited throughout the manuscript and appendices (e.g. mgcv and BORUTA)

We now cite the references for all packages used in R, including random forests.

L197: missing period

Changed

L198-L215: this ¶ is almost fully replicated in Appx 1 and could probably be summarized for concision in the main text

We had previous comments from other reviewers that this paragraph was a novelty in our modelling approach so we opt to continue to include.

L227: “the ‘local feature importance’ [or] the ‘situational importance’ of”

Changed

L229: replace “which is known as...” with a normal acronym definition

We replaced this phrase with the following on revised lines starting 236 *“To do so, we approximate Shapley values defined in game coalition theory⁴⁶ called SHapley Additive exPlanations (SHAP) when applied to explain machine learning predictions”*

L230-233: cut, move the pointer to the appendix elsewhere and any otherwise uncited XAI papers to L234

Changed

L234: consider adding a one-sentence summary of XAI in general here (or somewhere earlier)

We now add the following to revised lines 240: *“XAI methods aim to explain why complex “black box” models made predictions at an observation level.”*

L302, 318, 332, L344: I don't believe these equation definitions are completely well defined. The use of \in to define a set (i.e. “binary expected dist'n \in ”) is not correct, and the Σ notations index over N and A while the subscript on SHAP is always i. In L332, it's not clear to me whether \cap is being used as set intersection or a logical OR. I generally recommend against using \cap unless it is clear from context that the operands are both sets and \cap is the set intersection operator. As an example of the notation (set builder notation) I recommend, I would write L302 as “binary expected distribution := $\{s_i \mid \forall i \in M \mid \sum_{j \in S} \text{SHAP}_{\{ij\}} > 0\}$ ”, where s_i is the ith subcatchment in the set of all subcatchments M and S is the set of all abiotic variables.”. These equations should also probably be numbered.

Thanks for the suggested improvements. We now use the set builder notation as suggested and do not define sets with \in . We define the binary distributions as suggested, with a small notational change to keep with the described anthropogenic (A) and natural (N) terms for our variables as used throughout the manuscript. We no longer use the \cap operator. We now number all equations.

Overall, I found this section (L289-371) to be far too wordy and recommend cutting it down significantly and focusing the text more on the interpretability of these dist'ns. My expectation is that the two halves (expected and shadow dist'ns) could be condensed into one section to avoid duplicated and wordy variable definitions and allow for cleaner storytelling.

We have now shortened this section as suggested which now runs from revised lines 297 to 333.

L307-317, L338-344: is the expected distribution a set? a value calculated globally across a species' distribution? or a summary value at each site/subcatchment? This isn't clear at all from the text or equation.

We now clearly state that the binary expected distribution is a set on revised lines 298 *“We define a binary expected distribution as the set of sites (here sub-catchments) inside the abiotic niche of a species”*. We now define the quantitative expected distribution as a property of the sites inside the set on line 307 *“We define a property of each site inside the set defined by the binary expected distribution”*.

L319: if you're writing equations out, I suggest writing the subequation for \hat{y} as well below the one in L318

As this is simply the mean, we state on revised lines 310 *“ \hat{y} is the model mean predicted habitat suitability across all sites”* to avoid expanding too many equations.

L320: the \rightarrow this

Changed

L322-L325: I don't fully understand this and missed it entirely on my first readthrough

On revising this paragraph, this sentence was not clear and added confusion, so we have removed it.

L327: species → species'

Changed

L344: I don't believe $f(x_i)$ is defined anywhere? I think this is the missing equation from L322-325, but it really should be defined concretely if you're giving equations for these

We now use y_i as this is clearer than defining the output of an undefined function. We state on revised line 329 "*where y_i is the site environmental suitability score*"

L568: "theory, that" → "theory: that"

Changed

Thank you once again to the authors and editors for the opportunity to participate in the review of this manuscript. The new version is a significant improvement and I believe a very promising piece of work.

Thanks for your positive feedback and support.

Reviewer #3 (Remarks to the Author):

I would like to thank the authors for addressing most of my comments and providing in-depth answers. Thank you in particular for the following modifications, explanations, and additional experiments:

- The rephrasing of the Introduction and inclusion of (i.) contextual motivations like the GBF, and (ii.) SDMs. Much, much better overall!
- The inclusion of considerations regarding collinearity.
- The experiment on unmodified vs. 0-1 standardised covariates.
- Explanations regarding the downsampling procedure of decision trees. There seems to have been a misunderstanding on my end; this is now clear (also applies to other areas regarding the discussion; last few minor comments on original version).

We are very grateful for your feedback and constructive input to our work.

Comments on rebuttal:

"We have modified our introduction to better highlight where the GBF framework is relevant for our work. We were not sure exactly where in the framework the reviewer is referring to, but re-read the framework with specific focus on targets 1-3."

Correct, your amendments touch more or less on what I had in mind. The framework outlines a number of headline and component indicators, often on a species or larger spatial level (i.e., global to country). As is expected from such aggregated measures, they do not highlight local issues enough. Hence, the GBF also contains supporting indicators that are designed to pick up details at a finer scale, which is precisely what your study could latch on.

For example, CBD/COP goal A ("integrity, connectivity, and resilience of all ecosystems") contains headline/component indicators like the red list or living planet index (aggregated measures), whereas complementary indicators go down to more granular themes like land cover, algae cover, etc. All of these have to be measured and understood at all levels. Since we will inevitably use correlative models, employing XAI methods is arguably the best way to go; since it all boils down to anthropogenic change, disentangling contributions by type and introducing concepts like the shadow distribution do so as well.

We took no action as saw this comment in strong agreement with our changes.

Remaining larger comments:

- "The issue of sampling biases highlighted here is only valid for non-probabilistic samples, namely SDMs build from presence-only data. The raised issue is therefore not present in our models because we only used presence-absence data from standardized surveys that record all species present (and absent/not detected) in a location."

I fundamentally disagree. It is very well possible to have sampling bias-related issues under systematic presence-absence surveys also, just to a lesser and different degree. You mention yourselves that largest rivers have received fewer surveys; this doesn't mean that biases have been cancelled out. Paraphrased: one can easily set up systematic and rigorous presence/absence surveys but fail to distribute survey locations across habitats evenly. Hence, the following sentence:

"Further, any issues relevant for presence-only SDMs, such as sampling biases, are also problematic for SHAP explanations of these models, and as such we encourage the use of presence-absence data from standardized surveys to build SDMs."

We do agree that our electrofishing sampling data is biased in certain unavoidable ways towards or against certain habitats and therefore species. We note that all surveys have this source of bias. We added the following to lines 697-700 of the revised discussion to highlight this source of bias:

"We also note that probabilistic presence-absence surveys can still contain biases because some sampling methodologies bias against difficult to sample habitats, here large rivers, which biases the amount of data available across habitat gradients."

The main point pertinent to our research is that there are major and important biases in using presence-only data that are circumvented using presence-absence data. Presence-only data does not control for sampling effort. In contrast presence-absence SDMs standardise sampling effort and assumes detection error is constant. This sampling effort bias we refer to having largely excluded by using presence-absence data when compared to presence-only data.

- p.8, lines 902ff: "To calculate the SHAP values exactly is extremely computationally challenging because the number of possible coalitions increases exponentially with the number of variables, and predictions from all possible combinations of variables must be calculated with and without the focal feature." – this, in principle, is correct.

We think this comment comes largely from a error on our part. In response to previous reviews, we revised throughout the manuscript the use of Shapley values (as defined mathematically) to SHAP values (the estimation method for Shapley values) in all cases – however in this case we made a mistake and should refer to Shapley values, not SHAP values. In their exact form, *Shapley* values as defined by Shapley (1958) are almost computationally impossible to calculate exactly, and therefore require approximation methods – this is what we refer to in this sentence, sorry for the confusion. We decided on reflection that this text isn't necessary, as it simply explains why *Shapley* values are computationally expensive – but the SHAP values of our models run in a few hours so were very feasible to run and scale.

However, you are using a tree-based model (random forest), and for those, a computationally tractable method that is exact, known as "TreeExplainer", has been presented (Lundberg, S.M., Erion, G., Chen, H., DeGrave, A., Prutkin, J.M., Nair, B., Katz, R., Himmelfarb, J., Bansal, N. and Lee, S.I., 2020. From local explanations to global understanding with explainable AI for trees. *Nature machine intelligence*, 2(1), pp.56-67). In principle, this begs the question why you used an old, expensive, approximative Monte Carlo method. Sure, you present your method as being model-agnostic, but this by itself is a bit tricky (you cannot apply these methods reliably to models like e.g. convolutional neural networks).

Realistically speaking, I do not expect major differences to emerge across these methods; also, not using that version, despite its objective advantages, is of course by no means a death knell to the paper. However, for a journal like this one, I cannot leave the topic unaddressed, especially since SHAP values play such a central role in the methodology and others may want to implement your proposition. As a minimum, TreeExplainer needs to be referenced and (with a few sentences) addressed. An even better outcome would be to have a comparison between the Monte Carlo approximation and TreeExplainer on a subset of the data. Do you think this would be possible?

Thanks for highlighting TreeExplainer as really excellent looking potential method to also calculate SHAP values for tree-based methods such as random forests. We were aware of this approach but applied a model-agnostic method in our work. As the reviewer also confirms, our understanding is that this approach has the main advantage of speed, rather than presenting a systematic bias that might affect our results or conclusions. We apologies for the previous confusion where we argued SHAP values as computationally limiting, when we actually meant *Shapley* values – which may have misled the reviewer to understand we had strong limitations in time for running our specific analysis. Time constraints were not a major bottleneck in our work, but we appreciate these could be in others work (hence the revised text highlighted below). Our decision to use model-agnostic approaches is important because it is more relevant to species distribution modelling where users are often interested in using ensembles of multiple statistical algorithms (but rarely convolutional neural networks as mentioned by the reviewer). So our use of model-agnostic SHAP estimation methods allows our approach to be expanded within our field better. Additionally, the number of features in speed tests of Lundberg et al. (2020) begins at 20 features (Supporting Figure 4C) which is far higher than the number often used in species distribution modelling. As such, the time constraints highlighted by the reviewer not particularly relevant to our field where > 20 features are very rarely used.

In summary, i) the main issue perceived was slow running of our analysis, ii) this issue was previously miscommunicated by us, ii) the types of models fitted by SDM modellers leads to SHAP value estimation that is likely (and we found) much faster than that tested in Lundberg

et al. (2020), iv) our field often uses multiple model algorithms so model-agnostic approaches are beneficial for us to use and highlight.

As suggested by the reviewer, we chose to highlight TreeExplainer as a potentially faster alternative. We now cite TreeExplainer so that interested readers can see the other additional benefits of this package (such as interaction effects, which we do not highlight as they are not a major focus of our manuscript). We now state in revised lines 273-275:

“Note, however, that multiple options exist for calculating local model explanations and model-specific faster alternatives for tree-based methods exist that are a better alternative for larger datasets with more features, such as “TreeExplainer” (Lundberg et al. 2020).”

Minor comments:

Unfortunately the line numbering system used by the reviewer was completely different to our submitted manuscript. We made our best efforts to find all the relevant phrases. We also note that our submitted version has all figures at >600dpi resolution so do not know why the images were low resolution for this reviewer.

- p.3, l.544: You may want to define the "SHAP" acronym here (first mention).

Changed

- p.3, l.549ff: I recommend rephrasing; questions like that don't look well in a paper in my view. Also, not sure about the novelty claim regarding that point.

We aim to highlight the novelty of the question rather than the aggregation of SHAPs, we now rephrase this question on lines 79-81: *“We also aggregate SHAP values to explore the novel question of how positive effects of natural factors (a species' abiotic niche) and negative effects of threats influence a species populations across their geographic range.”*

- p.4, Fig.1: nice new figure, but again of low resolution and problematic colour ramps/combinations. You will likely be asked by the journal to improve that (also was a comment in my original review that remained unanswered).

We assume this is an issue with uploading/downloading across web platforms and the received document by the reviewers. All figures input to the original word document at >600dpi and the viewed proof had all images at high quality.

- p.7, l.712f: Thank you for adding this explanation, but you perhaps want to move it down not to disrupt the natural flow of information (CV first, then the final model).

Changed

- p.8, l.874ff: Again, just an ordering issue: you start by stating that you "provide a detailed explanation of SHAP values in Appendix 3", then follow it up with a (higher level) description right after. I would move that sentence about the supplementary explanations after line 891.

Moved as suggested

- p.11, l.1173-1191: this is neat idea, but it in particular requires discussing limitations and pitfalls of XAI methods (we're "playing" with SHAP values here in a simulated causal way). You may want to refer to this part again in the limitations section at the end of the discussion as an example of where to be cautious (cf. p.22, l.1893ff).

We now add the following to revised lines 667-672: *“We caution that in estimating shadow distributions, the choice of adjustment to “reference” state should be carefully explored. Furthermore, we note that other XAI tools such as counterfactual explanations could provide additional insights to the potential impact of alleviating threats, or further exploring scenario building by modifying the feature space which is common practice in biodiversity projections.”*

- p.19, l.1701ff: I like this part a lot. One aspect that would make it complete as an argument could be to add examples of how to measure human disturbances for other taxa, even if just in a parenthesis.

We appreciate the comment, but unfortunately, we do not know which line specifically is referred we suppose this is a relatively minor comment as each specific system will have different threats, stressors and datasets that best measure these.

- p.22, l.1883ff: Thank you also for that passage; great as well – apart from one detail (lines 1885f): SHAP values are not only "not inherently robust to correlated features", they require special treatment unless features are fully independent (Lundberg, S.M. and Lee, S.I., 2017. A unified approach to interpreting model predictions. *Advances in neural information processing systems*, 30).

We now cite this reference in this sentence and highlight it as an important paper for further discussion on this topic of the criteria for valid inference from SHAP values. Revised lines 658-661 now read:

“It should be noted that SHAP values are not inherently robust to correlated features, and we needed to check multi-collinearity of variables, decorrelate variables that exhibit (non-linear) dependence, and remove variables that exhibited biologically implausible relationships (see Lundberg and Lee (2017) for further discussion).”

- p.22, l.1898f: It's more as you have put it in the beginning of the paper: XAI methods describe both feature importance and model decision in a confused way.

Noted but we do not see a specific suggestion so left the sentence as it stands.

- p.23, l.1964ff: Thanks, this is better.

Thanks for the feedback.